



# Retrieval algorithm for OClO from TROPOMI by Differential Optical Absorption Spectroscopy

Jānis Puķīte[1], Christian Borger[1], Steffen Dörner[1], Myojeong Gu[1], Udo Frieß[2], Andreas Carlos Maier[3], Carl-Fredrik Enell[4,*], Uwe Raffalksi[4], Andreas Richter[3], and Thomas Wagner[1]

[1]Max Planck Institute for Chemistry, Mainz
[2]Institute of Environmental Physics, Heidelberg
[3]Institute of Environmental Physics, Bremen
[4]Swedish Institute of Space Physics, Kiruna
[*]now at EISCAT Scientific Association, Kiruna

**Correspondence:** Jānis Puķīte (janis.pukite@mpic.de)

**Abstract.** The TROPOspheric Monitoring Instrument (TROPOMI) is a UV-VIS-NIR-SWIR instrument on board of Sentinel-5P satellite developed for monitoring the Earth's atmosphere. It was launched on 13 October 2017 in a near polar orbit. It measures spectrally resolved earthshine radiances at an unprecedented spatial resolution of around 3.5x7.2 km² (3.5x5.6 km² starting from 6 Aug 2019) (near nadir) with a total swath width of ∼2600 km on the Earth's surface providing daily global
coverage. From the measured spectra high resolved trace gas distributions can be retrieved by means of differential optical absorption spectroscopy (DOAS).

Chlorine dioxide (OClO) is a by-product of the ozone depleting halogen chemistry in the stratosphere. Although being rapidly photolysed at low solar zenith angles (SZAs) it plays an important role as an indicator of the chlorine activation in polar regions during polar winter and spring at twilight conditions because of the nearly linear dependence of its formation to
chlorine oxide (ClO).

Here we present a new retrieval algorithm of the slant column densities (SCDs) of chlorine dioxide (OClO) by DOAS. To achieve a substantially improved accuracy, which is especially important for OClO observations, accounting for absorber and pseudo absorber structures in optical depth even of the order of $10^{-4}$ is important. Therefore in comparison to existing retrievals, we include several additional fit parameters accounting for spectral effects like the temperature dependency of the
Ring effect and Ring absorption effects, higher order term for the OClO SCD dependency on wavelength and account for the BrO absorption.

We investigate the performance of different retrieval settings by an error analysis with respect to random variations, large scale systematic variations as function of solar zenith angle and also more localised systematic variations by a novel application of an autocorrelation analysis.

The retrieved TROPOMI OClO SCDs show a very good agreement with ground based zenith sky measurements and are correlated well with preliminary data of the opeartional TROPOMI OClO retrieval algorithm currently being developed as part of ESA's S5p+I project.



## 1 Introduction

It is well known that catalytic halogen chemistry causes large depletion of ozone in polar regions in spring (WMO, 2018).
In particular, $Cl_2$ is released in large amounts by heterogeneous reaction of $ClONO_2$ and HCl on polar stratospheric clouds (PSCs). Once the air mass with $Cl_2$ becomes irradiated by sunlight, $Cl_2$ is subsequently photolysed to atomic Cl (Solomon et al., 1986). Atomic Cl can result also from other reactions like between $ClONO_2$ and liquid or solid phase $H_2O$ and subsequent photolysis of the produced HOCl or other reactions as very recently pointed out by Nakajima et al. (2020). Atomic Cl in turn reacts with ozone (Stolarski and Cicerone, 1974). Because the resulting ClO (with or without involvement of BrO) is returned to atomic Cl (Molina and Molina, 1987; McElroy et al., 1986) by further reactions, a very effective ozone depletion process takes place. Furthermore, chlorine dioxide (OClO) is a possible outcome of a reaction between ClO and BrO (Sander and Friedl, 1989):

$$ClO + BrO \rightarrow Br + OClO \qquad (R1)$$

The dominant loss mechanism for atmospheric OClO is its very rapid photolysis (Solomon et al., 1990):

$$OClO + h\nu \rightarrow ClO + O \qquad (R2)$$

which results in a null cycle with respect to ozone loss by recycling odd oxygen. Thus, OClO can be used as an indicator for halogen chemistry because of the nearly linear dependence of OClO formation to ClO and BrO concentrations (Schiller and Wahner, 1996) at high solar zenith angles where the photolysis is slow enough to provide OClO abundances above the detection limit for the passive scattered light UV/VIS measurements (Solomon et al., 1987). Measurements of OClO in the context of the passive scattered light UV/VIS measurements became of special interest when Solomon et al. (1987) measured it first in Antarctica. OClO has well-structured absorption features in the near UV (Wahner et al., 1987) which can be detected by Differential Optical Absorption Spectroscopy (DOAS) (Platt and Stutz, 2008).

First satellite retrievals of OClO were enabled by measurements of the GOME-1 instrument launched in 1995, consequently leading to many studies investigating the polar stratospheric chlorine activation (Burrows et al., 1999; Wagner et al., 2001, 2002b; Kühl et al., 2004a, b; Richter et al., 2005). Later also measurements by SCIAMACHY, OSIRIS, OMI or GOME-2 were available for analysis (Kühl et al., 2006; Krecl et al., 2006; Kühl et al., 2008; Pukīte et al., 2008; Oetjen et al., 2011; Hommel et al., 2014).

The TROPOspheric Monitoring Instrument (TROPOMI) is a UV-VIS-NIR-SWIR nadir viewing instrument on board of the Sentinel-5P satellite developed for monitoring the Earth's atmosphere (Veefkind et al., 2012). It was launched on 13 October 2017 in a near polar orbit. It measures spectrally resolved earthshine radiances at an unprecedented spatial resolution of around 3.5x7.2 km² (near nadir) at a high signal-to-noise ratio. The measurements are performed via a pushbroom spectrometer consisting of 450 detector rows. It has a total swath width of $\sim$2600 km on the Earth's surface providing daily global coverage.





The spatial resolution has been further increased to 3.5x5.6 km² (near nadir) starting from 6 Aug 2019 (Rozemeijer and Kleipool, 2019).

In this paper we present a new retrieval algorithm for slant column densities (SCDs) of OClO by applying the DOAS method to TROPOMI measurements. The article is structured as follows: in Sect. 2 we present the TROPOMI OClO retrieval algorithm, while in Sect. 3 we compare the results with ground based zenith sky measurements obtained during polar Arctic winters in Kiruna, Sweden and Antarctic winters in Neumayer, Antarctica. Sect. 4 correlates the retrieved OClO SCDs with preliminary data of the TROPOMI OClO retrieval algorithm currently being developed as part of ESA's S5p+I project. Finally, Sect. 5

draws some conclusions.

## 2   Retrieval algorithm

The principle of the DOAS method (Platt and Stutz, 2008) is based on the application of the Beer – Lambert law by performing a least squares fit to best scale the contributions of the fit parameters to the measured optical depth (logarithmic normalized intensity):

$$\ln\frac{I(\lambda)}{I_0(\lambda)} = -\sum_i S_i \cdot \sigma_i(\lambda) + P \qquad (1)$$

    where $I$ and $I_0$ are radiances of the measurement and reference (or Fraunhofer) spectra, respectively. $S_i$ is a scaling factor of constituent $i$ and $\sigma_i(\lambda)$ is its wavelength dependent cross-section. $P$ is a broad band spectral contribution due to light scattering approximated by a polynomial. In the first place, scaling factors $S_i$ describe trace gas slant column densities (SCDs), habitually interpreted as number densities of trace gases integrated along their effective light paths. Besides that, also other parameters,

scaling linearly in optical depth space, that as approximations account for additional spectral features with corresponding pseudo cross-sections, can and, depending on circumstances, should be included in the fit. They can include, but are not limited to, the Ring effect (Wagner et al., 2009), offset correction, shift and stretch (Rozanov et al., 2011; Beirle et al., 2013), tilt effect (Rozanov et al., 2011; Lampel et al., 2017), polarization corrections (McLinden et al., 2002; Kühl et al., 2006), changes in the instantaneous spectral response function (ISRF) with respect to the ISRF obtained during calibration (Beirle et

al., 2017) or higher order contributions to account for non-linearities in absorption and SCD dependency on scattering (Puķīte et al., 2010; Puķīte and Wagner, 2016).

    SCDs obtained from measurements in the limb mode (vertical scanning of the atmosphere with instrument pointing above the Earth's horizon as done by OSIRIS and SCIAMACHY) can be converted to vertical OClO concentration profiles (Krecl et al., 2006; Kühl et al., 2008) or even 2D vertical concentration fields along the orbit by means of a tomographic approach

(Puķīte et al., 2008). For nadir measurements, SCDs can be converted to vertical column densities (VCDs, vertical integrals of number density profiles) by applying radiative transfer modelling (Solomon et al., 1987; Wagner et al., 2001; Kühl et al., 2004a). However, it is often preferred to omit this step for nadir measurements of OClO mainly because of two reasons (Kühl et al., 2004a): First, for the radiative transfer simulations, the atmospheric state must be known well (especially at





high SZAs ($> \sim 85°$) which are of special interest for OClO) including the OClO profile which is highly variable due to the large photochemical reactivity (Solomon et al., 1990). Second, the OClO VCD at these high SZAs cannot be interpreted as a measurement of chlorine activation level , since they still depend on the photolysis rate of OClO, which in turn strongly depends on the intensity of solar illumination and thereby also on the SZA (Kühl et al., 2004a). In other words the calculation of the VCD at a given location would necessarily require a-priori constrains about the concentration variability along the light path. Therefore also this study limits the retrieval to SCDs.

## 2.1 Retrieval settings

The retrieval of the OClO SCDs is performed by a universal fit routine which was originally developed for the TROPOMI water vapour retrieval (Borger et al., 2020). Retrieval parameters for OClO are provided in Table 1.

Before the application of the DOAS method, a spectral calibration is performed for each of the TROPOMI rows separately to obtain measurement wavelength grids and instrumental spectral response functions (ISRFs). The calibration includes a non-linear least-squares fit in intensity space with respect to a high resolution solar spectrum (Kurucz et al., 1984) for wavelength alignment and slit function determination approximated by an asymmetric Super-Gaussian function as described by Beirle et al. (2017). The calibration is done for the reference spectrum ($I_0$) for which in our application we take a daily mean of the normalized Earth-shine measurements within a solar zenith angle range $60 - 65°$ as defined in Appendix A1.2. The normalization is done for each spectrum by its maximum spectral intensity value within the evaluated spectral range. The usage of an Earth-shine reference is motivated by the need to reduce detector-related effects (e.g. Wagner et al., 2001; Kühl et al., 2006), in particular the detector striping, as well as the amplitude of the Ring effect which are both more prominent when direct Sun measurements are used. The use of an Earthshine reference does not hinder the interpretation of the retrieved OClO data because no OClO is expected to be found at these low SZAs. It is hence expected that the mean of the retrieved OClO slant column densities in the reference region is zero. The use of the normalized spectra (Appendix A1.2) for the calculation of the daily mean ensures that also spectral features that are not related to OClO but correlate with its cross-section are not producing an artificial offset. The effect for this application is however negligible.

The DOAS analysis is performed within a fit window from 363 to 390.5nm. Absorption cross-sections of OClO at 213 K (Kromminga et al., 2003), $NO_2$ at 220K (Vandaele et al., 1998), $O_3$ at 223K (Serdyuchenko et al., 2014) and $O_4$ at 293 K (Thalman and Volkamer, 2013) are included in the fit. $P$ is approximated by a 5th order polynomial. For the convolution of the cross-sections to the instrument spectral resolution, an intensity weighted convolution (to account for $I_0$ correction) is applied as described in Appendix A2 (Eq. A9).

In the fit we account also for the intensity offset, shift and stretch (Beirle et al., 2013) as well as ISRF parameter changes (Beirle et al., 2017). The Ring effect is accounted for by Ring spectra calculated at two temperatures (280K and 210K) from the reference spectrum and each scaled or unscaled with $\lambda^4$ according to Wagner et al. (2009).

Also a $\lambda \cdot \sigma_{OClO}$ term according to the approach of Puķīte et al. (2010); Puķīte and Wagner (2016) (convolved with intensity weighted convolution, Eq. A11) is included in the fit. Besides its physical meaning to account for the broadband wavelength dependency of the OClO SCD due to the change of light path distribution in the atmosphere which is large at high SZAs, this



**Table 1.** Retrieval settings for OClO SCD fit.

| Parameter | Description |
| --- | --- |
| Input data | TROPOMI Band 3 spectral radiances (320 – 405 nm) |
| Calibration | Non-linear least squares fit for wavelength alignment and Super Gaussian ISRF approximation (Beirle et al., 2017) |
| Fit window | 363–390.5 nm |
| Reference spectrum | Daily mean Earth-shine spectrum within SZA range 60 – 65°, detector resolved |
| Polynomial | 5th order |
| Absorption cross-sections | OClO, 213 K (Kromminga et al., 2003) |
| | $NO_2$, 220K (Vandaele et al., 1998) |
| | $O_3$, 223K (Serdyuchenko et al., 2014) |
| | $O_4$, 293 K (Thalman and Volkamer, 2013) |
| Ring effect | 4 Ring spectra (2x2) at 280 and 210K, scaled or unscaled with $\lambda^4$ |
| Ring effect on absorption | Ring $NO_2$ cross-section (see Sect. A3) |
| Higher order absorption corrections (Puķīte and Wagner, 2016) | $\lambda \cdot \sigma_{OClO}$ |
| Additional pseudo absorbers | intensity offset (terms $1/I_0$, $\lambda/I_0$, $\lambda^2/I_0$) |
| | shift and stretch (Beirle et al., 2013) |
| | ISRF changes (Beirle et al., 2017) |
| BrO absorption correction | subtraction of BrO absorption, retrieved in another fit window (Warnach et al., 2019), from the measurement before the fit (see Sect. A4) |

parameter gives more flexibility to better account for the parameter cross-correlation problem because one can calculate the final OClO SCD at a wavelength which is less affected by systematic errors caused by the imperfection of the model.

The retrieved OClO SCDs are obtained from the fitted quantities according to the formalism in Puķīte and Wagner (2016) by:

$$S_{OClO}(\lambda) = S_{OClO,\sigma} + \lambda S_{OClO,\lambda\sigma} \qquad (2)$$





where $S_{OClO,\sigma}$ is the fit result corresponding to the OClO cross-section and $S_{OClO,\lambda\sigma}$ to the $\lambda \cdot \sigma_{OClO}$ term. $\lambda = 379$ nm is selected for the evaluation.

Additionally, a cross-section is added to approximate the impact due to the Ring effect on the $NO_2$ absorption. This $NO_2$ Ring absorption spectra term accounts for the first order $NO_2$ absorption contribution of the Raman scattered light. The definition is provided in Appendix A3. Last but not least it turned out to be advantageous to correct the measured spectra prior to the OClO DOAS fit for the BrO absorption because we found an interference of the retrieved OClO SCDs towards BrO. However, inclusion of BrO cross-section as an additional fit parameter led to even larger systematic errors. Therefore a BrO correction

(with the BrO absorption determined in a different spectral range) is applied before the OClO retrieval on the measured spectra as described in Appendix A4.

## 2.2   Retrieval performance and spatial data binning

Uncertainties in passive scattered light DOAS measurements depend on retrieval noise, the accuracy of the removal of Fraunhofer bands, the absorption cross-sections, unexplained spectral structures and effects of the radiative transfer (Platt and Stutz,

2008). For the OClO cross-section by Wahner et al. (1987) systematic errors ($\leq 8\%$) could be due to the absolute calibration (Wagner et al., 2001). For the cross-section used in this study, Kromminga et al. (2003) stated that their cross-sections agree within those of Wahner et al. (1987) within a similar error range. Thus, we assume a similar error ($\sim 10\%$). In the following the retrieval performance is investigated by means of a statistical analysis. The impact of the retrieval settings is studied in Appendix B motivating the retrieval setup we introduced in Sect. 2.1.

### 2.2.1   Systematic and random errors

In order to determine the contribution of systematic errors we make use of the fact that OClO occurs only for limited time periods and areas. Thus, we investigate the retrieved OClO SCDs for scenes where no OClO absorption is expected. This allows to investigate the retrieval performance by means of a statistical analysis. The left panel in Fig. 1 shows daily means of retrieved OClO SCDs for three days in each hemisphere binned on a SZA grid with a bin size of $0.2°$. The days are selected

to represent different atmospheric conditions. 25 Aug 2018 for NH and 25 Dec 2018 for SH are summer days in the respective hemispheres where no enhanced OClO SCDs are expected. 25 Nov 2018 (NH) and 25 Apr 2019 (SH) are early winter days with temperatures not yet having dropped below the PSC forming temperature, thus again no enhanced OClO SCDs are expected. 25 Dec 2018 (NH) is an example for a day with enhanced OClO concentrations present, the level however is low as it is the case in this winter. 25 Aug 2018 (SH) shows typical OClO values for the Antarctic winter atmosphere with a very strong

chlorine activation. The right panel in Fig. 1 illustrates the standard deviation of the OClO SCDs, a measure we use for the mean random error.

     Both the retrieved mean OClO SCDs as well as their standard deviations show a pronounced increase with increasing SZA. The increase of the standard deviations can be expected due to a lower signal to noise ratio at twilight. Considering the days for which no enhanced OClO SCDs are expected, the daily mean OClO (or systematic offset with respect to the expected zero)

is largest for SH being around $1 \times 10^{13}$ cm$^{-2}$ where the offset sign is different between December and August.





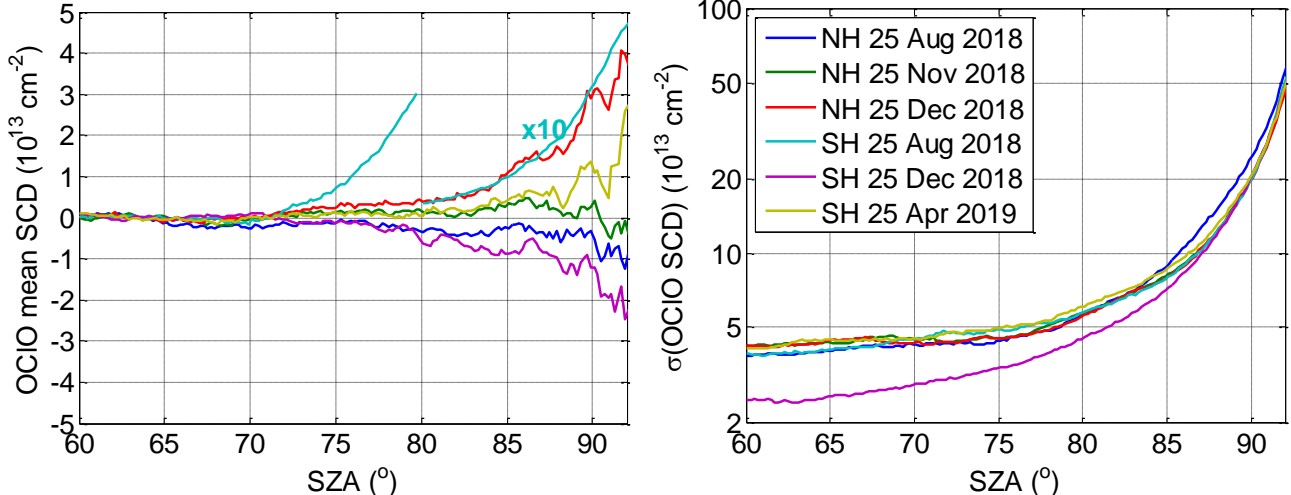

**Figure 1.** Left: OClO mean SCDs resolved on 0.2° SZA grid for 3 days indicated in the legend on the left plot in different seasons for each hemisphere. Note that the SCDs as plotted for 25 Aug 2018 in the SH must be multiplied by the factor of 10 for SZAs above 80°. Right: same but standard deviation of OClO SCDs.

.

The standard deviation at the same time is very similar between the hemispheres and the different seasons. Being $4 \times 10^{13}$ cm$^{-2}$ at low SZA of 60° it reaches $\sim 2 \times 10^{14}$ cm$^{-2}$ at SZA of 90°. One exception is the smaller value of $2.5 \times 10^{13}$ cm$^{-2}$ at SZA of 60° for SH in 25 Dec 2018 where most of the measurements within the 60–85° SZA region are located over the Antarctic ice improving the signal to noise ratio.

**2.2.2 Autocorrelation**

In an ideal case each measurement at one time and location is independent from all the others. In practice there are different effects that cause a correlation between measurements. In the previous section it was already demonstrated that the systematic errors depend on parameters like SZA and season. Besides those there are additional sources of systematic errors that are more localized. From a purely instrumental point of view such sources would be caused by the point spread function partly

overlapping sensitivity areas of nearby measurements. A correlation could also be caused by other effects like e.g. the memory effect or striping effect. While in the case of a striping effect all measurements made by the same detector are correlated to some degree, the effects caused by the point spread function and the memory effect would affect all nearby measurements. Another kind of systematic error is related to variations within the observed scene which are caused by the variation of atmospheric conditions (like e.g. cloud) or surface properties. There are two kinds of impacts of the observed scene on the retrieved OClO

SCDs: First, it causes localized systematic patterns (patches) in the 2D distribution indicating a correlation between individual measurements within a certain proximity. Second, the correlation between different rows is observed due to systematic patterns in the observed scene at the region where the Earthshine reference is taken.



The autocorrelation function is a tool in signal as well as image analysis which quantifies the correlation between data points at different distances apart from each other (Chatfield, 2003; Jähne, 2005). It has the benefit that the autocorrelation

patterns can be analysed even at large random error levels (like it is the case for the retrieved OClO SCDs from the individual TROPOMI measurements). To our knowledge, so far there has been no application of an autocorrelation analysis reported in the literature for the systematic error analysis in satellite trace gas measurements. For our purposes it provides a handy tool to quantitatively compare the performance of different retrieval settings with respect to systematic errors as done in Appendix B. This subsection only introduces the concept and applies the calculations to the standard scenario.

The autocorrelation coefficient $\rho$ for OClO SCDs $S$ for the lags $\Delta i$ and $\Delta j$ in along and across track dimension with respect to individual pixels $i$ and $j$ is:

$$\rho_{\Delta i, \Delta j} = \frac{C_{\Delta i, \Delta j}}{(I \cdot J)\sigma^2} = \frac{\sum_{i,j} (S_{i,j} - \overline{S})(S_{i+\Delta i, j+\Delta j} - \overline{S})}{(I \cdot J)\sigma^2} \tag{3}$$

with $C$ being the auto covariance, $\overline{S}$ – the mean SCD of the subset, $\sigma$ – the standard deviation of the subset and $I \cdot J$ the size of the dataset. For practical reasons, $C$ is calculated by a Fast Fourier transformation invoking the Wiener–Khinchin theorem:

$$C(\Delta i, \Delta j) = \mathcal{F}^{-1}(|\mathcal{F}(S(i,j))|^2) \tag{4}$$

Since the variation of OClO along the orbit is not a completely spatially stationary process (given e.g. that its mean and standard deviation depend on SZA), we investigate a subset consisting of all across track measurements with their across-track mean SZA being between 60° and 75° for the days for which no enhanced OClO is expected, introduced in Sect. 2.2.1. Figure 2 presents the obtained correlograms. For all days the autocorrelation coefficients are quite similar and show a maximum (except

the self-correlation at 0 pixel lags being unity by definition) at lags of 1 pixel in either dimension. The distinct single correlation peak at small lag values decreases quickly towards lags of 1–2 pixels in both dimensions by a factor of two with respect to the values at lag 1. We could speculate that since this short scale correlation occurs always, it is mostly caused by instrumental factors (e.g. by the spatial response function, memory effect). The maximum value of below 4% shall be interpreted in a relative sense because it depends on the selection of the subset region and the sample standard deviation. The correlation at

intermediate lag distances can be interpreted as caused by retrieval artefacts with respect to the scene parameters.

While at even larger lags the autocorrelation reaches zero in the across track direction, it stays enhanced in the along track direction and is distinctively larger when the across-track lag is 0. These distinctively larger values for the lags along track at the across-track lag of zero are caused by the detector striping effect. The correlation between nearby stripes at large lags along track is likely caused by structural patterns in the observed scene at the region where the Earthshine reference has been

obtained. According to Fig. 2, the correlation between nearby detectors seems to be smaller for 25 December in the SH, which could be caused by a less structured impact of the reference spectra across track than for other days as above ice the cloud effect plays a smaller role.

**Figure 2.** Left: Mean autocorrelation coefficients of orbital slices with across track scans taken at SZAs between 60° and 75° for days without enhanced OClO as indicated in the respective colourplots. On the left and bottom of each colourplot the coefficients are plotted as line plots for the lag across track of 0 and 1 pixel and lag along track of 0, respectively. The values for both lags being zero are excluded (it is unity by definition). Note that the axis limits are selected to be in the same range as for the plots for the sensitivity studies with different retrieval parameters in Appendix B

.

In summary, the contribution of local scale systematic errors that can be detected by the autocorrelation analysis (the mean relation of one measurement to another in a statistical sense) is only a few percent, thus it is not a dominating error source

at the original instrument resolution. The investigation of autocorrelations however is still important because for binned measurements (see next subsection), the local scale systematic errors become important because of a lower variance of the gridded



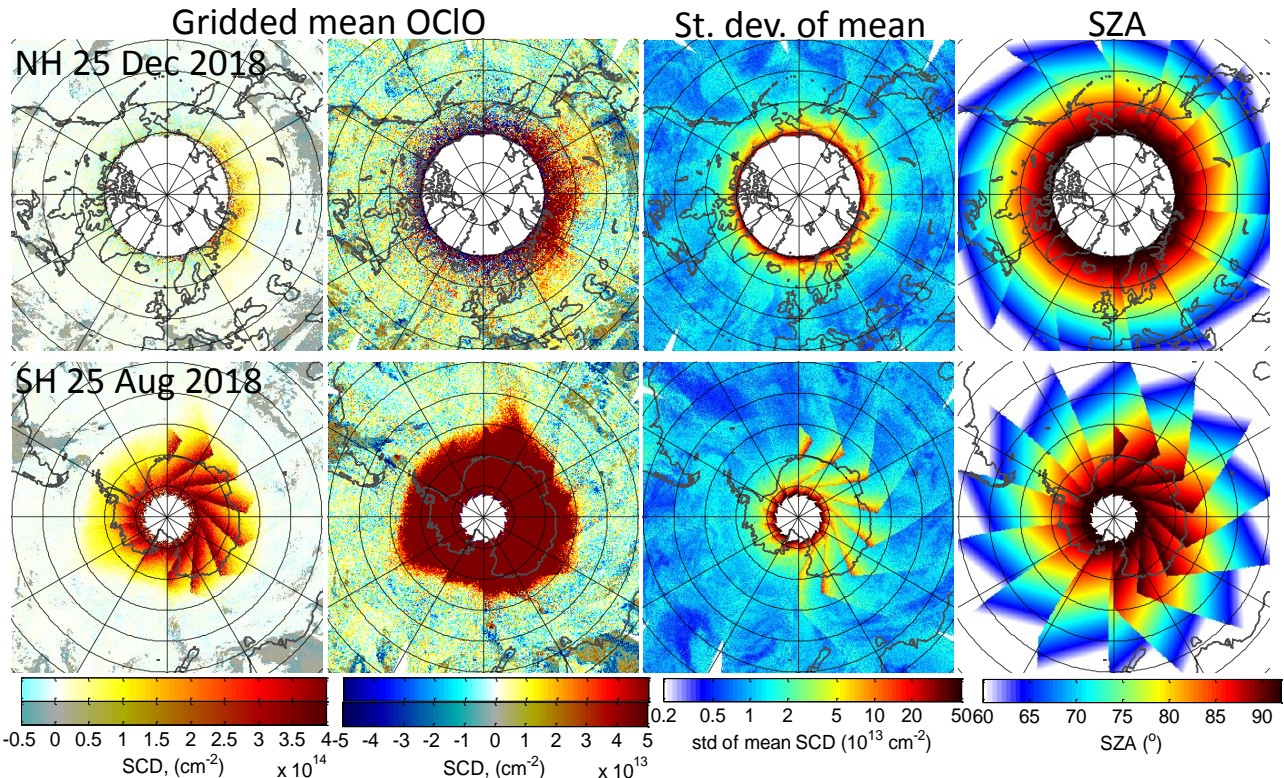

**Figure 3.** Left plots: OClO SCDs for 25 Dec 2018 in the NH (top) and 25 Aug 2018 (bottom) in the SH binned on a 0.2°×0.2° grid in terms of an equidistant in latitude coordinate projection. Areas with cloud fraction (CF) below 5% are shaded. Second column from left: same results but with a different colour scale. Second from right: standard deviation of the binned mean. Right: SZA of the plotted data.

.

mean. Thus, the autocorrelation analysis provides an important contribution when evaluating the retrieval performance with different retrieval settings in the sensitivity studies in Appendix B.

### 2.2.3 Binning

Similarly as for the autocorrelation analysis, the large random errors (Fig. 1, right), having the same magnitude as the OClO SCDs even for strong chlorine activation levels, make measurements hardly interpretable by eye at the TROPOMI resolution. Therefore we apply a spatial binning. Figure 3 shows OClO SCDs for 25 Dec 2018 in the NH and 25 Aug 2018 in the SH binned on a 0.2°×0.2° grid in an equidistant in latitude coordinate projection (corresponding to an area of roughly 20×20 km$^2$). Note that in the left columns the same results are shown, but with colour scales either to illustrate the OClO SCD variation in 215 the atmosphere or for the purpose of illustrating systematic features, compare Fig. 1). The figure also illustrates the standard deviation of the gridded mean (second column from right) and the SZA (right column). The gridded data show areas with increased OClO SCD values distinct from the very smooth background. This 'real' OClO signal for the gridded data remains





pronounced even for rather low OClO levels as for 25 Dec 2018 in the NH because the standard deviation of the binned data is only around $5\times10^{13}$ cm$^{-2}$ at a SZA of 90°. It is worth mentioning that a value of $5\times10^{13}$ cm$^{-2}$ is often referred to as a

value for the OClO detection limit in the literature for GOME-1 and SCIAMACHY (Wagner et al., 2001; Oetjen et al., 2011). At lower SZAs the standard deviation of the mean decreases quickly by a factor of 10. This allows to identify even localized systematic features whose existence were deduced from the spatial cross-correlation study in the autocorrelation study (see Sect. 2.2.2). They typically do not deviate from zero by more than $\pm1\times10^{13}$ cm$^{-2}$ except for areas with very low cloud fraction (CF) where the deviation can be by a factor of two or three larger. Unfortunately, the performed sensitivity studies

have not provided an explanation for this dependency. We could speculate that for clear sky cases the lower signal to noise ratio, larger effects of the spectral straylight, imperfections in the detector linearity or the spectral polarization sensitivity of the instrument could play a larger role. To mark such cases, areas with CF below 5% for SZAs below 75° are shadowed in the figure.

Because OClO is highly variable, it is not possible to provide a general value for a typical relative error. Thus, we limit the

error estimation to a few values for possible OClO SCD levels. The relative error of OClO SCD gridded data is about 15% for a SCD of $5\times10^{14}$ at SZA of 90$^o$. This estimate is obtained by adding the squares of the relevant errors (cross-section error 10%, random error $5\times10^{13}$ cm$^{-2}$ and assuming the systematic error (offset magnitude) of $2\times10^{13}$ cm$^{-2}$). For a SCD of $2\times10^{14}$ this translates to an error of 30%. For lower SZAs where both the expected levels of the retrieved OClO and the random error are substantially smaller, the relative error at a SZA 85$^o$ for an OClO SCD of $5\times10^{13}$ is estimated as 50%.

The detection limit which we determine as the SCD value that corresponds to a relative error of 100% is about $6\times10^{13}$ cm$^{-2}$ and $2.5\times10^{13}$ cm$^{-2}$ at SZA of 90$^o$ and 85$^o$, respectively.

# 3 Comparison with ground-based zenith sky measurements

Although a nadir viewing satellite instrument like TROPOMI and a ground-based zenith sky instrument are viewing in opposite directions (one downwards from space and another upwards from the Earth's surface), the absorption by OClO at high SZAs

is taking place in the stratosphere, which, before the light is scattered into the instrument's field of view, is crossed by the light under the same slant angle for both instruments, i.e. is probing nearly the same air mass. Thus it is expected that also nearly the same photochemical variations along the light path take place. We compare OClO SCDs retrieved from TROPOMI measurements with those obtained by ground based zenith sky DOAS instruments at Kiruna (Gottschalk, 2013; Gu, 2019), Sweden (67.84°N, 20.41°E) and at Neumayer station (Frieß et al., 2005) in Antarctica (70.64°S, 8.26°W). For the comparison,

the OClO SCDs retrieved for all TROPOMI pixels within 100 km radius around the ground stations are averaged. To filter for outliers TROPOMI pixels with fit errors above $10\times10^{14}$ cm$^{-2}$ are excluded. Also cases with less than 100 TROPOMI pixels within that area are excluded from further processing. For the zenith-sky instruments, SCDs within the SZA range of $\pm0.5°$ around TROPOMI SZA are averaged.

**Figure 4.** OClO SCDs measured at Kiruna (67.84°N, 20.41°E) by the zenith sky DOAS instrument (black line). Their diurnal variation reflects their dependency in particular on SZA. The zenith sky measurements at the TROPOMI SZAs are marked by 'x'. Collocated TROPOMI measurements are indicated by 'o'. The SZAs of these measurements are colour-coded while the radius of the circles indicates with the number of TROPOMI pixels within the selected area (in range 100 – ~1500).

.

## 3.1 Kiruna

The Zenith sky DOAS instrument at Kiruna measures scattered sunlight between 300 and 400 nm with a spectral resolution of about 0.6 nm and approximately 10 times finer wavelength sampling. OClO SCDs from Kiruna used for this comparison are retrieved in fit window of 372 – 392 nm. Cross-sections for OClO (Kromminga et al., 2003) at 213 K, ozone (Bogumil et al., 2003) at 223K, $O_4$ (Thalman and Volkamer, 2013) at 273 K and $NO_2$ (Vandaele et al., 1998) at 220 K are considered. The intensity offset is considered by $1/I_0$ and $\lambda/I_0$ terms. Also 4 Ring spectra (at temperatures of 213 and 263 K, scaled and not

scaled with $\lambda^4$) and a $\lambda \cdot \sigma_{OClO}$ term (Puķīte and Wagner, 2016) are included in the fit. As a Fraunhofer reference an OClO



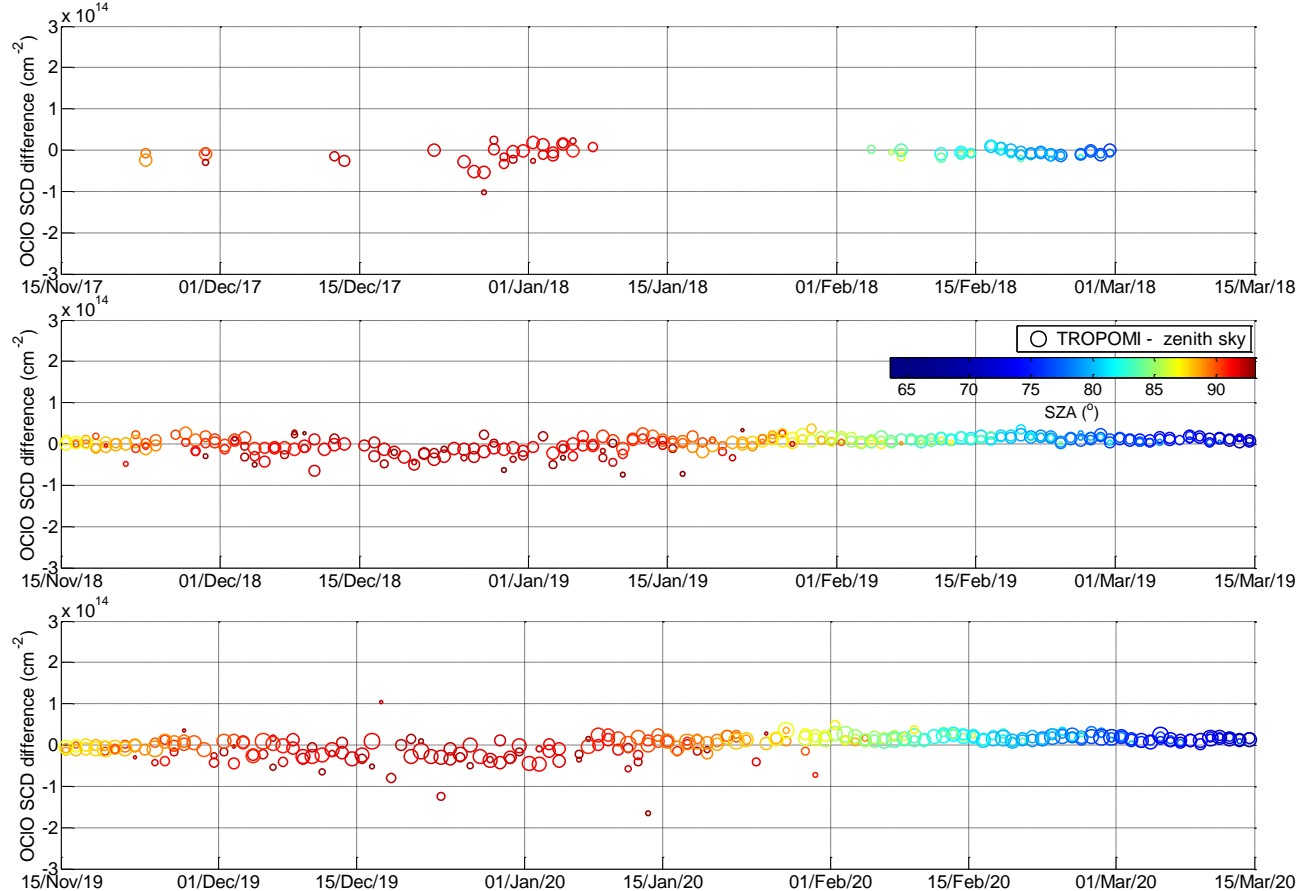

**Figure 5.** Difference between the collocated TROPOMI OClO SCD measurements and the ground based measurements in Kiruna for three different winter. Colour coding and radius of the circles as in Fig. 4.

.

free spectrum, obtained before the winter in October at about 80° SZA at a day when Kiruna was outside the polar vortex, is taken.

The time series comparing both data sets are shown in Fig. 4 where the collocated TROPOMI measurements are indicated as circles, and zenith sky measurements as crosses. Both are coloured according to the SZA. The circle radius also scales with the available TROPOMI pixel numbers within the collocation area ranging from 100 – approx. 1500. Figure 5 shows the difference between both datasets. Generally very good agreement can be found not exceeding $\pm5\times10^{13}$ cm$^{-2}$ for SZA at and below 90°. Larger discrepancies (mostly within $\sim1\times10^{14}$ cm$^{-2}$) appear at very high SZAs, where the scatter can be larger especially for comparisons with a low number of averaged TROPOMI pixels. This can also be seen in Fig. 6 where on the left side the difference is plotted between the collocated TROPOMI and zenith sky DOAS measurements as function of the SZA (x-axis). The time of the measurement is provided by the colourscale. For SZAs below 90° a systematic offset of about $1\times10^{13}$





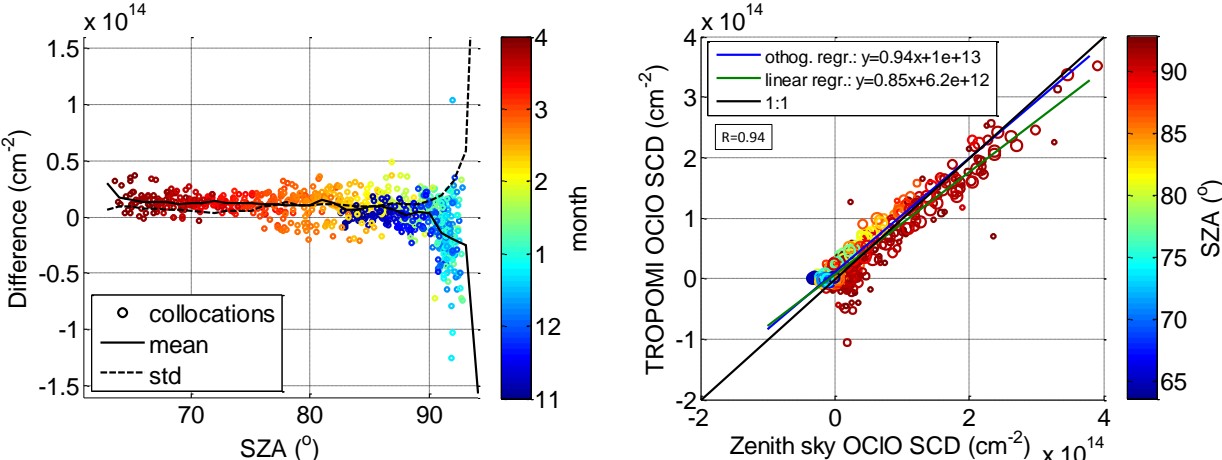

**Figure 6.** Left: Difference between the colocated TROPOMI measurements and zenith sky DOAS measurements as function of SZA (x-axis) and the time of the measurement (colourscale). The mean and standard deviation of the data binned in a 1° SZA grid are indicated by the black solid and dashed line, respectively. Right: Correlation plot between the TROPOMI and zenith sky measurements. The SZAs of these measurements are colour-coded while the radius of the circles indicates the number of TROPOMI pixels within the collocation area. Orthogonal and linear regression lines between both datasets and their equations are provided in the legend.

cm$^{-2}$ is found with a standard deviation that does not exceed this offset value. Close to the SZA of 90° the mean offset crosses zero and becomes negative at larger SZAs with standard deviation of about $4\times10^{13}$ cm$^{-2}$. Figure 6, right, shows a scatter plot between the TROPOMI and zenith sky data. The data are again coloured with respect to the SZA and the collocated TROPOMI pixel numbers. The plot includes two regression lines: for a linear regression as well as an orthogonal regression with the latter

being reported to be more adequate for independent datasets (Cantrell, 2008). Both regression results indicate good agreement. Nevertheless the slope for the orthogonal regression (0.94) is much closer to unity than for the linear regression (0.85). The offset parameters are $1.0\times10^{13}$ cm$^{-2}$ and $6.2\times10^{12}$ cm$^{-2}$, respectively. The correlation coefficient between both datasets is 0.94.

Last but not least it is worth to mention that the additional Ring spectra and the $\lambda\cdot\sigma_{OClO}$ term in the retrieval of Kiruna OClO

SCDs retrieval improved the comparison with the satellite measurements dataset considerably. In Appendix C the comparison as in this subsection is presented but with Ring terms at only one temperature and without the $\lambda\cdot\sigma_{OClO}$ term.

## 3.2 Neumayer

The UV channel of the MAX-DOAS instrument at Neumayer station in Antarctica (Ferlemann et al., 2000; Frieß et al., 2001, 2005) measures scattered sunlight between 320 and 420 nm with a spectral resolution of 0.5 nm (full width half maximum)

by a 1024 element photodiode array. Note that for the OClO analysis only the zenith measurements are used. OClO SCDs for Neumayer are retrieved in the wavelength range of 364 – 391 nm. The fit settings include a 4th order polynomial, cross-sections for OClO (Kromminga et al., 2003) at 213 K, ozone (Bogumil et al., 2003) at 223K and 293K, NO$_2$ (Vandaele et al., 1998) at





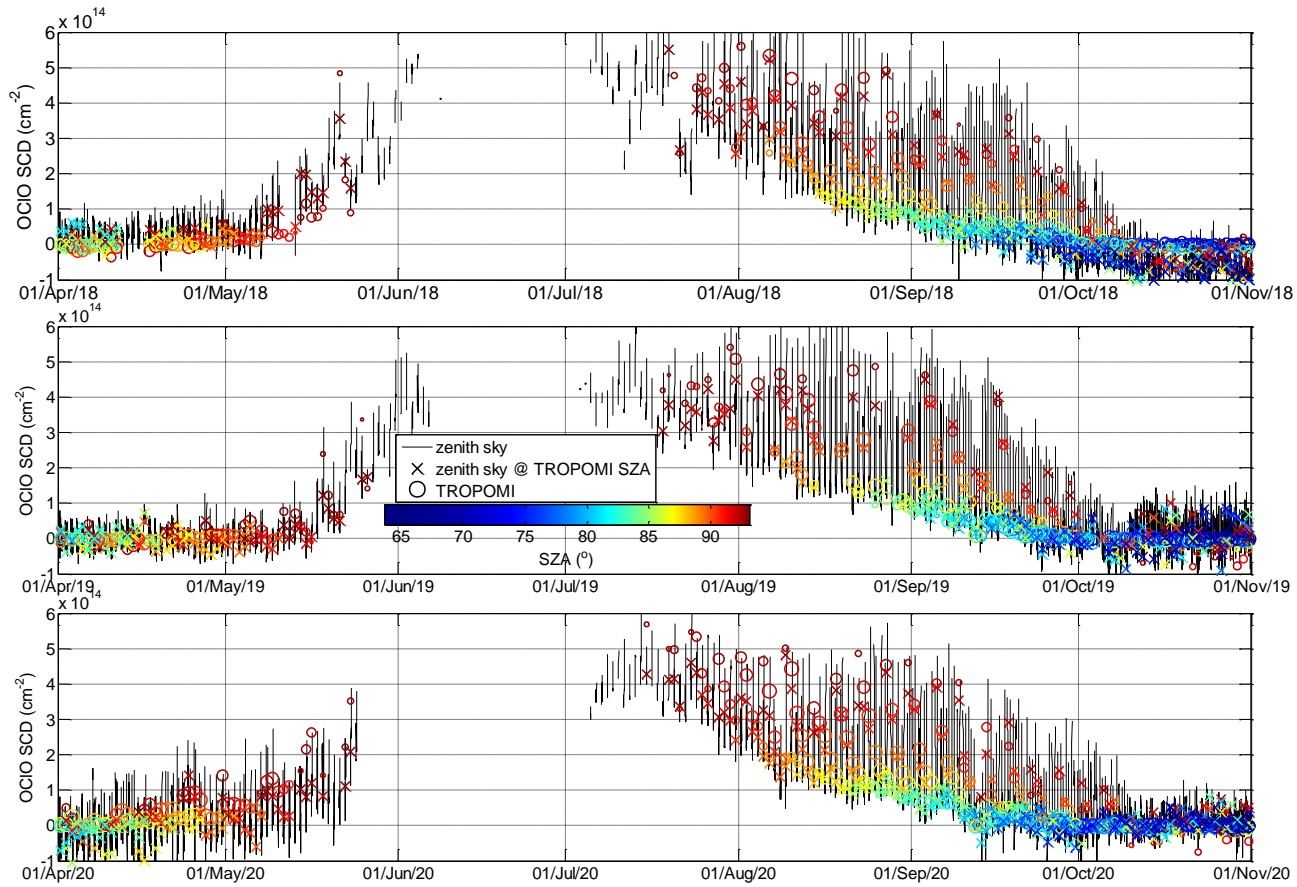

**Figure 7.** Same as Fig. 4 but for zenith sky OClO SCDs measured at Neumayer (70.64°S, 8.26°W).

220K and 298K, O4 (Hermans) at 298 K, and BrO (Wilmouth et al., 1999) at 228K. Also 2 Ring spectra (scaled and not scaled with $\lambda^4$) and intensity offset ($1/I_0$ term) are fitted. For the Fraunhofer reference, spectra in sunlit atmosphere outside the polar
vortex are recorded.

The time series shown in Fig. 7 is similar to that shown in the Fig. 4 for the Kiruna measurements. Also the differences are provided in the same way in Fig. 8. Also for the Neumayer data good agreement is observed. The differences in most cases do not exceed $\pm10\times10^{13}$ cm$^{-2}$ for SZA at and below 90° with a seasonal drift from local autumn until spring in the range of $\pm5\times10^{13}$ cm$^{-2}$. The drift varies from year to year: For the polar winter 2018 the differences range from being mostly negative
in April and May to mostly positive in September and October. For the winter 2019 the differences are scattered around zero both at the beginning and end of the season. However an opposite trend is found for the winter 2020: the differences are mostly positive in April and May and become more negative in October. For all winters mostly positive differences are observed at the end of July and beginning of August. This can also be seen in the left plot in Fig. 9 where the difference between the collocated



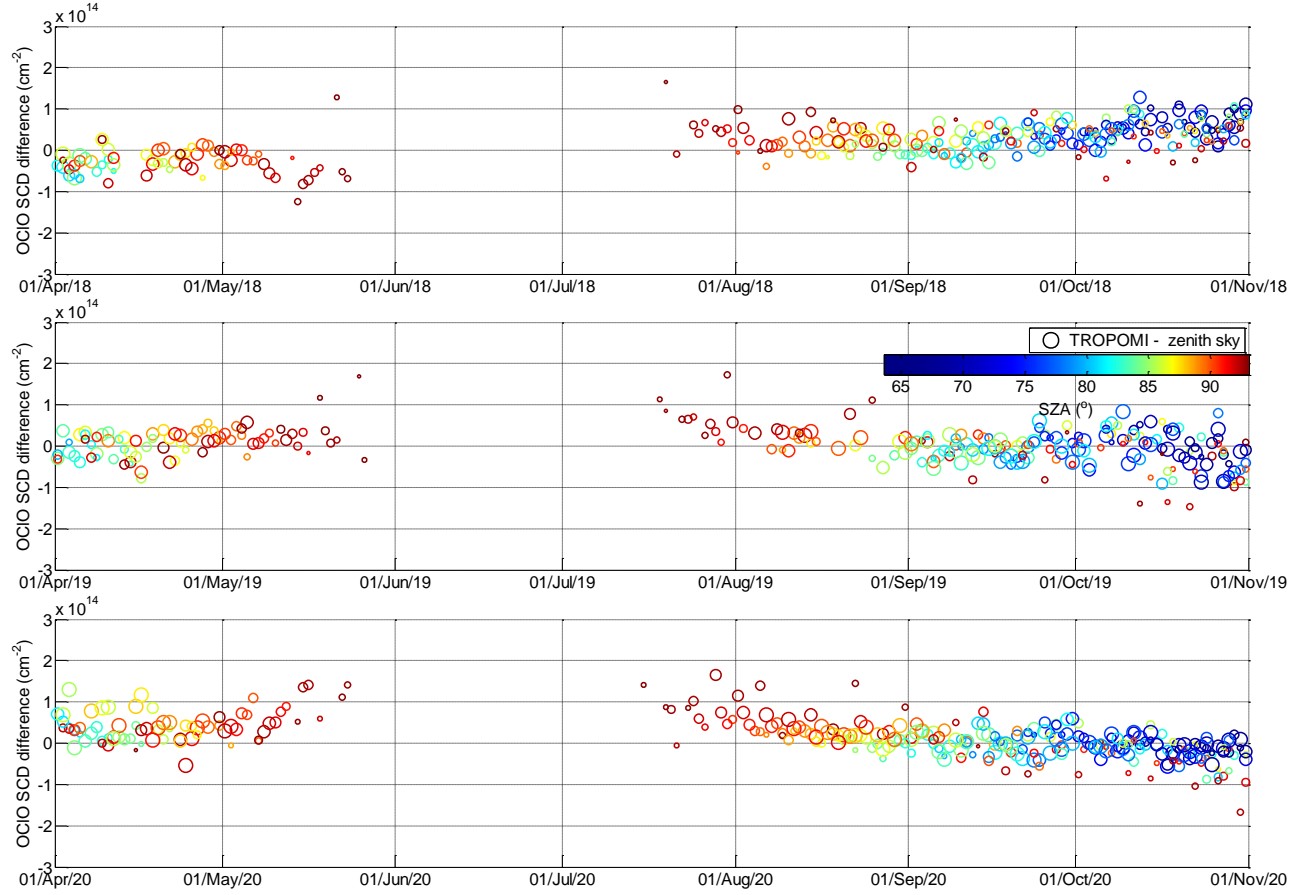

**Figure 8.** Same as Fig. 5 but for zenith sky OClO SCDs measured at Neumayer.

.

TROPOMI and zenith sky DOAS measurements is plotted as function of the SZA (with the time of the measurements indicated
by the colourscale). At low SZA which appear in April and also in September and November, the mean difference is in the
range between 0 and $3 \times 10^{13}$ cm$^{-2}$, where the scatter reflects both the day to day variability and the differences between the
different winters. The offset in July and beginning of August causes increased mean differences at a level of 2 and $4 \times 10^{13}$
cm$^{-2}$ for SZAs above 87°. At SZAs above 90° the scatter increases and also the differences between measurements in May and
end of July-August show a different offset. The standard deviation of the differences is rather constant with SZA being around
$4$–$5 \times 10^{13}$ cm$^{-2}$ for SZAs around 90° and below. Figure 9, right, shows a scatter plot between the TROPOMI and zenith sky
data. The linear regression provides slope parameter of 1.1, while for the orthogonal regression it is 0.98. The offset is 0 and
$1.3 \times 10^{13}$ cm$^{-2}$, respectively. The correlation coefficient (0.96) between both datasets is very similar to that for Kiruna (0.94).





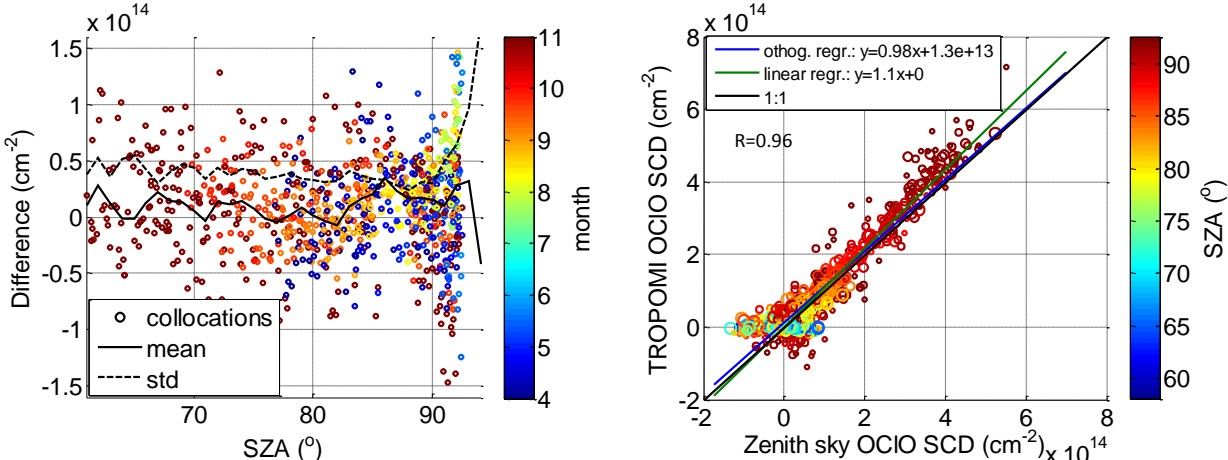

**Figure 9.** Same as Fig. 6 but with zenith sky OClO SCDs measured at Neumayer (70.64°S, 8.26°W).

## 4  Comparison with preliminary S5p+I OClO product

In the framework of the Sentinel5P Innovation activity (S5p+I) an operational OClO product from TROPOMI is under devel-
opment. The current state of the algorithm (v0.95) is described in the Algorithm Theoretical Basis document (Mayer et al.,
2020) in detail. Here just the basic settings are listed: Retrieval applies a solar irradiance spectrum as Fraunhofer reference and
uses a fit window of 345 – 389 nm. Cross-sections of OClO (Kromminga et al., 2003) at 213 K, $NO_2$ (Vandaele1998) at 220
K, ozone (Serdyuchenko et al., 2014) at 223 K and 243 K, $O_4$ (Thalman and Volkamer, 2013) at 293K, Ring (Vountas et al.,
1998) and intensity offset ($1/I_0$) terms. Also two empirical cross-sections derived from mean residuals are used to account for
contributions not explained by the already considered fit parameters.

We compare the retrieved OClO from our study with the preliminary S5p+I OClO SCD mean data within 89°<SZA<91°
range.

Figure 10 plots the time series of the S5p+I OClO SCDs together with those from this study for the polar winters 2018/2019
and 2019/2020 in the NH, and 2018, 2019 and 2020 in the SH. While the overall shape agrees very well, an offset between
both datasets is observed for low OClO levels where the S5p+I data have higher values. As will be discussed in the next section
(Sect. **??**), OClO is not expected at the beginning of November (NH) or April (SH) because of still warm temperatures. Thus
values systematically above zero should likely be an artefact. At very high OClO levels the agreement becomes better. This
behaviour is confirmed in the correlation plots in Fig. 11. A very high correlation is obtained with correlation coefficient being
0.990 in the NH and 0.996 in the SH. The orthogonal regression has offset of $-5.4 \times 10^{13}$ and $-3.6 \times 10^{13}$ cm$^{-2}$, and the slope
is 1.10 and 1.07, respectively.

The slope is likely caused by the wavelength dependency of the OClO SCDs: For the interpretation we assume the same
Gaussian profile shape as for BrO for the simulations in Fig. A1. The obtained ratio at the SZA of 90° between SCDs simulated
at 380 and 340 nm is 1.35 – 1.9 (depending on profile peak altitude). Keeping in mind that the OClO SCDs for our study are



**Figure 10.** Timeseries of TROPOMI mean OClO SCDs within 89°<SZA<91° obtained within this study and the preliminary data from the operational S5p+I data for polar winters (from top to bottom) 2018/2019 and 2019/2020 in the NH, and 2018, 2019 and 2020 in the SH.

obtained at 379 nm, and assuming that the effective wavelength for the S5p+I is in the middle of their fit window (i.e. at 367
nm), we obtain a ratio in the range of 1.085–1.17 (broadly consistent with the observed slope).

The offset could be related to still uncompensated higher order effects in the current version of the S5p+I OClO fit, as the consideration of the wavelength dependency of fit parameters becomes more challenging in larger fit windows. Also the application of the solar irradiance instead of the earthshine spectrum as Fraunhofer reference might have an influence.



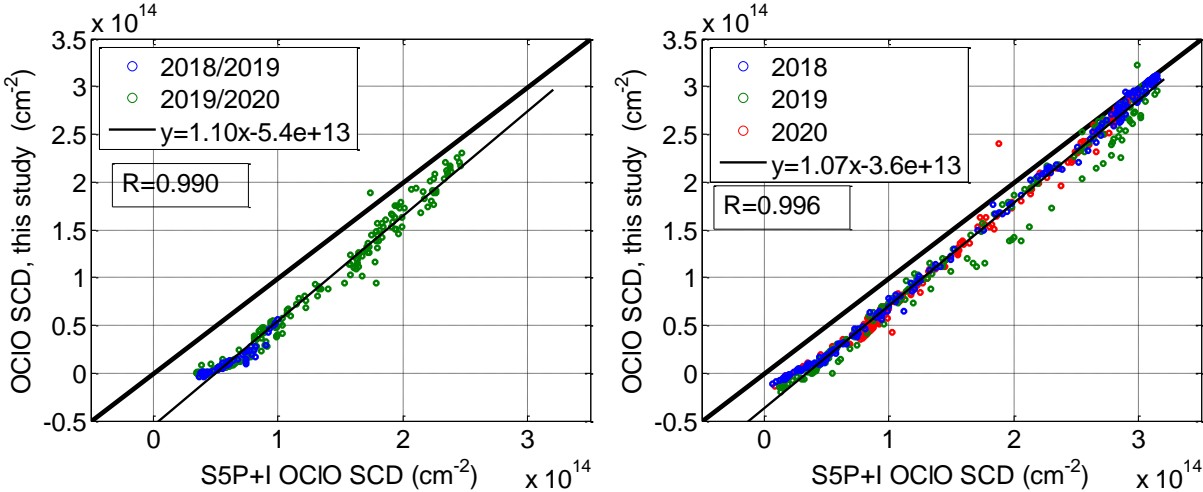

**Figure 11.** Correlation plots between the time series shown in Fig. 10 left for the NH winters, right for the SH winters.

## 5    Conclusions

We developed a novel retrieval algorithm of OClO SCDs from the TROPOMI instrument on Sentinel-5P. To achieve the challeging high accuracy (and low detection limit), which is especially important for OClO observations, accounting for absorber and pseudo absorber structures in optical depth even of the order of $10^{-4}$ is important. Therefore in comparison to existing retrievals, we include several additional fit parameters accounting for spectral effects like the temperature dependency of the Ring effect and Ring absorption effects. The analysis also considers higher order terms (Puķīte and Wagner, 2016) and the

BrO absorption contribution. Including these terms improves the retrieval results especially for low OClO SCDs. The typical random error is around $4 \times 10^{13}$ cm$^{-2}$ at low SZA ($60°$) and reaches $\sim 2 \times 10^{14}$ cm$^{-2}$ at an SZA of $90°$. Thus for the interpretation of the data, averaging of individual measurements is needed to decrease the random error towards the typical levels of OClO SCDs. In this study we average the individual measurements on a horizontal grid of about $20 \times 20$ km$^2$ (averaging about 20—25 individual measurements) which is well suited for measurements in the stratosphere.

For the gridded OClO SCDs, the random uncertainty is typically $0.5 - 1 \times 10^{13}$ cm$^{-2}$ at low SZA and $5 \times 10^{13}$ cm$^{-2}$ at SZA around $90°$. Also, the systematic errors are in this range mostly not exceeding $1–2 \times 10^{13}$ cm$^{-2}$. Thus the detection limit is about $0.5–1 \times 10^{14}$ cm$^{-2}$ at SZA of $90^o$ which is similar to the detection limits of earlier instruments.

We investigated the performance of different retrieval settings by an error analysis with respect to random variations, large scale systematic variations as function of SZA and also more localised systematic variations by a novel application of an

autocorrelation analysis.

The retrieved dataset agrees very well with zenith sky measurements in both polar regions (at Kiruna and Neumayer station) with slopes and correlation coefficients near unity. The larger absolute differences between individual TROPOMI and Neumayer datapoints are compensated by much higher absolute OClO SCDs at Neumayer. The use of similar settings for the





spectral analysis for the Kiruna zenith sky measurements as those used for the TROPOMI analysis with an optimized treat-
ment of the temperature dependency of the Ring effect and also includes an OClO cross-section 'lambda term' significantly
improved the agreement practicaly removing the year to year and seasonal variability in the difference between the TROPOMI
and zenith sky datasets.

A nearly perfect correlation (correlation coefficient being practically unity) is obtained with the comparison to the prelim-
inary data of the operational S5P+I retrieval algorithm. In the S5P+I data however a systematic positive offset is found. Here
we can only speculate that this offset might be caused by higher order effects, which are probably more important in the
larger fit window of the S5P+I retrieval. A slope slightly different from unity can largely be explained by the differences in the
wavelength regions used for the DOAS fit in both analyses.

*Data availability.* Data are available upon request

## Appendix A: Definition of some retrieval concepts and settings

## A1 Calculation of the Earthshine reference spectra

### A1.1 Mean Earthshine reference

When an Earthshine reference is used for the DOAS analysis, the measurements in a selected reference region are averaged
obtaining a mean radiance reference spectrum $I_{ref}(\lambda)$:

$$I_{ref}(\lambda) = \frac{1}{n} \sum I_i(\lambda) e^{-\tau_i(\lambda)} \tag{A1}$$

where $I_i(\lambda)e^{-\tau_i(\lambda)}$ indicate the individual measured spectra with $I_i(\lambda)$ and $\tau_i(\lambda)$ scaling linearly in the intensity and optical
density domains, respectively.

In DOAS, the logarithm of Eq. A1 is used for the normalization of individual measurements. Expanding it in a Taylor series
up to the first order with respect to $\tau_i(\lambda)$ one obtains:

$$ln(I_{ref}(\lambda)) = ln(\frac{1}{n} \sum I_i(\lambda) e^{-\tau_i(\lambda)}) \approx ln\frac{1}{n} \sum I_i(\lambda) - \frac{\sum \tau_i(\lambda) I_i(\lambda)}{\sum I_i(\lambda)} \tag{A2}$$

The second term on the right side indicates that the features $\tau_i(\lambda)$ in the mean Earthshine reference spectrum is weighted
with the intensities of the contributing measurements. Thus, such a reference spectrum in the DOAS analysis generally would
lead to an offset for the fitted parameters even in the reference region. Even in the case that there is no expected absorption of
a particular absorber, the potential errors and the incompleteness of the representation of the atmospheric state by the DOAS
model can induce an offset.





### A1.2 Weighted mean Earthshine reference with inverse radiances weights

To avoid this problem we normalize the individual measurements in the reference region before averaging:

$$I_{ref}(\lambda) = \frac{\sum \frac{I_i(\lambda)e^{-\tau_i(\lambda)}}{I_i(\lambda')}}{\sum \frac{1}{I_i(\lambda')}} \tag{A3}$$

Intensity $I_i(\lambda')$ is selected at a distinct wavelength $\lambda'$ within measurement spectrum $I_i(\lambda)$.

Expanding the logarithm of Eq. A3 in a Taylor series up to the first order with respect to $\tau_i(\lambda)$ one obtains:

$$ln(I_{ref}(\lambda)) = ln\frac{\sum \frac{I_i(\lambda)e^{-\tau_i(\lambda)}}{I_i(\lambda')}}{\sum \frac{1}{I_i(\lambda')}} \approx ln\frac{\sum \frac{I_i(\lambda)}{I_i(\lambda')}}{\sum \frac{1}{I_i(\lambda')}} - \frac{\sum \tau_i(\lambda)\frac{I_i(\lambda)}{I_i(\lambda')}}{\sum \frac{I_i(\lambda)}{I_i(\lambda')}} \tag{A4}$$

Since $I_i'$ is nearly proportional to all intensities in the spectrum $I_i$, there is practically no effective weighting for $\tau_i$ in the second term on the right side. Thus using a weighted spectrum (Eq. A3) as a Fraunhofer reference spectrum in the DOAS analysis should not lead to an offset in the mean values of the fitted parameters in the reference region.

### A2 Intensity weighted convolution of cross-sections

The so called $I_0$ correction is applied to account for the so called absorption filling-in effect when convoluting absorber cross-sections to the instrument resolution in DOAS applications (Wagner et al., 2002a; Aliwell et al., 2002). An $I_0$ corrected cross-section $\sigma_{LR}$ for the low resolution (LR) domain is calculated as follows:

$$\sigma_{LR} = -ln\frac{\int I_0(\lambda)e^{-S\sigma_{HR}(\lambda)}K(\lambda_0 - \lambda)d\lambda}{\int I_0(\lambda)K(\lambda_0 - \lambda)d\lambda}/S \tag{A5}$$

where $I_0(\lambda)$ is the HR sun spectrum, $K$ is the convolution kernel (i.e. ISRF), and $S$ is the SCD. The result of the equation is twofold: first, it weights the absorptions at the individual wavelengths with the intensities of $I_0$, second, it considers nonlinear effects of absorptions in the exponent. It follows that the equation, however, is valid for a single absorber: In such a case the optical depth of the absorption is solely due to this one absorber and can be described in the LR domain by the logarithmic ratio of the convolved spectra with absorption and without absorption:

$$\tau_{LR} = S\sigma_{LR} = -ln\frac{\int I_0(\lambda)e^{-S\sigma_{HR}(\lambda)}K(\lambda_0 - \lambda)d\lambda}{\int I_0(\lambda)K(\lambda_0 - \lambda)d\lambda} \tag{A6}$$

The effective cross-section in the LR domain is just the ratio of $\tau_{LR}$ and $S$, the last term is assumed to be constant and as such can be retrieved by the fit.





The problem is to deal with a situation when the assumption of a single absorber is not fulfilled, in particular if there are more than one absorber in the scene or the SCD varies with wavelength. In such a case the absorption optical depth is:

$$\tau_{LR}(\lambda) = -ln\frac{\int I_0(\lambda)e^{-\sum_i S_i(\lambda)\sigma_{HR,i}(\lambda)}K(\lambda_0 - \lambda)d\lambda}{\int I_0(\lambda)K(\lambda_0 - \lambda)d\lambda} \tag{A7}$$

400 Now the absorption optical depth cannot be normalized by a constant SCD of a single absorber. Moreover the absorptions $S_i(\lambda)\sigma_{HR,i}(\lambda)$ of the individual trace gases $i$ have multiplicative effects, thus in principle they cannot be convoluted separately and their LR optical depths cannot be separated as would be needed for a correct definition of distinct LR cross-sections.

**A2.1 Weak absorption formalism**

Assuming that the absorption is weak enough, a Taylor series expansion can be applied (i.e. $e^x \approx 1 + x$ for $x$ close to 0), we 405 can however allow us to state:

$$\tau_{LR}(\lambda) \approx -ln\frac{\int I_0(\lambda)(1 - \sum_i S_i(\lambda)\sigma_{HR,i}(\lambda))K(\lambda_0 - \lambda)d\lambda}{\int I_0(\lambda)K(\lambda_0 - \lambda)d\lambda} \tag{A8}$$

Now the summands can be integrated separately. Assuming for that also the SCDs are constant, the $I_0$ corrected cross-sections for a weak absorption limit can be defined:

$$\sigma_{LR,i} = \frac{\int I_0(\lambda)\sigma_{HR,i}(\lambda)K(\lambda_0 - \lambda)d\lambda}{\int I_0(\lambda)K(\lambda_0 - \lambda)d\lambda} \tag{A9}$$

410 In cases where it is necessary to account for the SCD variation with wavelength, $S_i$ can be expanded in a Taylor series with respect to the wavelength and the cross-section (Puķīte et al., 2010; Puķīte and Wagner, 2016):

$$S_i(\lambda) = S_{0,i} + S_{\lambda,i}\lambda + S_{\sigma,i}\sigma_{HR,i}(\lambda) \tag{A10}$$

Putting this expression in Eq. A8 one just needs to convolve in addition to Eq. A9 also the products of $\lambda\sigma_{HR,i}(\lambda)$ and $\sigma_{HR,i}(\lambda)^2$ in order to obtain the $I_0$ corrected cross-sections for these higher order terms:

$$(\sigma\lambda)_{LR,i} = \frac{\int I_0(\lambda)\sigma_{HR}(\lambda)\lambda K(\lambda_0 - \lambda)d\lambda}{\int I_0(\lambda)K(\lambda_0 - \lambda)d\lambda} \tag{A11}$$

$$(\sigma^2)_{LR,i} = \frac{\int I_0(\lambda)\sigma_{HR}(\lambda)^2 K(\lambda_0 - \lambda)d\lambda}{\int I_0(\lambda)K(\lambda_0 - \lambda)d\lambda} \tag{A12}$$

By this intensity weighted convolution of the cross-sections, the first aim of the $I_0$ correction (weights the absorptions at individual wavelengths with the intensities of $I_0$) is fulfilled. For the application to the OClO SCD retrieval in this study, the 420 intensity weighted convolution (i.e. Eq. A9) is applied for all trace gas cross-sections. The cross-section wavelength term (Eq. A11) is used for the $\lambda \cdot \sigma_{OClO}$ term. Because of the weak absorption the cross-section square term (Eq. A12) is not applied.





### A2.2 Strong absorption assumption

Although being not relevant for the OClO SCD retrieval, we take for the sake of completeness a short excurse to show that also the second aim of the $I_0$ correction (i.e. accounting for the nonlinear contribution to the absorption filling effect) can be considered in a similar manner by just the intensity weighted convolution of the higher order terms.

For a case with a strong absorption when the usage of the Taylor series square term is needed and thus also an absorption nonlinearity effect in the $I_0$ correction can be expected, we need just to explore the higher order Taylor series expansion terms for the exponent of Eq. (A7). Consequently, Eq. A8 can be written in a general form:

$$\tau_{LR}(\lambda) \approx -ln\frac{\int I_0(\lambda)(1-\sum_g S_g\sigma_{HR,g}(\lambda))K(\lambda_0-\lambda)d\lambda}{\int I_0(\lambda)K(\lambda_0-\lambda)d\lambda} \tag{A13}$$

where $S_g\sigma_{HR,g}$ describes the absorption contribution of a particular term $g$ obtained by the Taylor series expansion (for the definition of the higher order DOAS terms please refer to Pukīte and Wagner (2016), in particular Sect. 3.3. therein).

The $I_0$ corrected cross-sections are then generally defined as (compare to Eq. A9)

$$\sigma_{LR,g} = \frac{\int I_0(\lambda)\sigma_{HR,g}(\lambda)K(\lambda_0-\lambda)d\lambda}{\int I_0(\lambda)K(\lambda_0-\lambda)d\lambda} \tag{A14}$$

These terms include both the cross-section wavelength term $(\sigma\lambda)_{LR,i}$ (Eq. A11) as well as the cross-section square term $(\sigma^2)_{LR,i}$ (Eq. A12). Thus it can be assumed that the intensity weighted convolution of the higher order terms implicitly accounts also for the non-linearity of the $I_0$ filling in.

### A3 Ring effect on NO$_2$ absorption

The Ring effect describes the so called "filling in" of solar Fraunhofer lines (Grainger and Ring, 1962) for scattered light measurements caused by the fact that part of light that undergoes inelastic (Raman) scattering. It results in a highly structured spectral fingerprint with respect to the direct Sun spectrum (see e.g. Wagner et al., 2009). Although the DOAS retrieval with an Earthshine spectrum as a Fraunhofer reference largely reduces the magnitude of the Ring effect, it still needs correction because of the different filling-in magnitudes between the measurement and the Fraunhofer reference.

In addition we found that it is necessary to account for the filling in of the NO$_2$ absorption in order to eliminate a pronounced systematic negative bias in the retrieved OClO SCDs at high SZAs, see Appendix B8. The absorption filling-in appears because a part of the absorption occurs before the Raman scattering event where light is crossing atmosphere at a different wavelength than after the scattering. This effect tends to reduce the differential trace gas absorption (smoothing effect) and seems to be not substantially reduced by the usage of the Earthshine reference in our retrieval algorithm because of significant differences in light paths of the inelastically scattered light between the measurement and the Fraunhofer reference.

Fish and Jones (1995) have first demonstrated the importance of the absorption filling where they estimated the underestimation in the retrieved NO$_2$ slant column from studies on simulated spectra. Van Roozendael et al. (2006) applied an iterative





correction in the ozone SCD retrieval by scaling of the effective slant column. More recently, Lerot et al. (2014) developed a semi-empirical formulation to calculate filling-in factors for their ozone retrieval, which, once multiplied to the elastically scattered radiances, correct them iteratively for the structures introduced by the inelastic scattering processes.

For our correction we just use the Taylor series expansion, as outlined below, of the Ring spectrum with respect to absorption
up to the first order and include the obtained first order Ring absorption term (for $NO_2$) as a free fit parameter directly in the fit, thus having a benefit that neither a radiative transfer modelling nor an iteration scheme are needed.

The inelastic scattering contribution at the detector wavelength $\lambda_0$ without absorption is defined as:

$$I_{inelastic}(\lambda_0) = \frac{1}{\int \sigma_{Raman}(\lambda, \lambda_0)d\lambda} \int R(\lambda)\sigma_{Raman}(\lambda, \lambda_0)d\lambda \tag{A15}$$

$\lambda$ is the incident wavelengths of the light entering the atmosphere and undergoing Raman scattering to the measurement
wavelength $\lambda_0$. $\sigma_{Raman}$ is the Raman line spectral cross-section, obtained according to Bussemer (1993).

While this Raman-scattered light travels through the atmosphere also absorption by trace gases takes place:

$$I_{inelastic,abs}(\lambda_0) = \frac{1}{\int \sigma_{Raman}(\lambda, \lambda_0)d\lambda} \int R(\lambda)e^{-(S_b\sigma_{abs}(\lambda) + S_a\sigma_{abs}(\lambda_0))}\sigma_{Raman}(\lambda, \lambda_0)d\lambda \tag{A16}$$

$S_b$ and $S_a$ describe parts of an absorber SCD (b)efore and (a)fter a Raman scattering event, respectively. $\sigma_{abs}$ is the cross-section of the relevant absorber. We consider here that just one Raman scattering event for individually contributing photon
paths takes place and assume that the contribution to the absorption filling in by light paths with more than a single Raman scattering event along one light path is negligible.

Elastically scattered contribution with absorption (assuming the same effective light path) is given as:

$$I_{elastic,abs}(\lambda_0) = R(\lambda_0)e^{-(S_b+S_a)\sigma_{abs}(\lambda_0)} \tag{A17}$$

The Ring spectrum (Bussemer, 1993; Wagner et al., 2009) is defined as the ratio between the inelastic and elastic terms:

$$\frac{I_{inelastic,abs}(\lambda_0)}{I_{elastic,abs}(\lambda_0)} = \frac{\int R(\lambda)e^{-(S_b\sigma_{abs}(\lambda) + S_a\sigma_{abs}(\lambda_0))}\sigma_{Raman}(\lambda, \lambda_0)d\lambda}{(\int \sigma_{Raman}(\lambda, \lambda_0)d\lambda)R(\lambda_0)e^{-(S_b+S_a)\sigma_{abs}(\lambda_0)}} \tag{A18}$$

After mathematical simplification of the expression we obtain:

$$\frac{I_{inelastic,abs}(\lambda_0)}{I_{elastic,abs}(\lambda_0)} = \frac{\int R(\lambda)e^{-S_b(\sigma_{abs}(\lambda) - \sigma_{abs}(\lambda_0))}\sigma_{Raman}(\lambda, \lambda_0)d\lambda}{R(\lambda_0)\int \sigma_{Raman}(\lambda, \lambda_0)d\lambda} \tag{A19}$$

Now the equation is linearized by expanding it in a Taylor series with respect to $S_b$:

$$\frac{I_{inelastic,abs}(\lambda_0)}{I_{elastic,abs}(\lambda_0)} \approx \frac{\int R(\lambda)\sigma_{Raman}(\lambda, \lambda_0)d\lambda}{R(\lambda_0)\int \sigma_{Raman}(\lambda, \lambda_0)d\lambda} - S_b\frac{\int R(\lambda)(\sigma_{abs}(\lambda) - \sigma_{abs}(\lambda_0))\sigma_{Raman}(\lambda, \lambda_0)d\lambda}{R(\lambda_0)\int \sigma_{Raman}(\lambda, \lambda_0)d\lambda} \tag{A20}$$





The first term on the right side is the classical Ring spectrum. The second term considers the Ring effect due to filling in of
trace gas absorptions. As it can be seen, this filling in depends both on the absorption cross-section and the intensity structures,
the product of whose are convoluted with the Raman scattering cross-section. In this way individual Ring absorption spectra
for any relevant absorber can be calculated and considered in the fit. In our retrieval it appeared to be relevant to consider a
Ring absorption spectrum just for $NO_2$. Here $S_b$ is the quantity obtained by the fit, the rest is the Ring absorption spectrum.

It is worth mentioning that for our application the Ring absorption spectrum is originally calculated on a high resolution grid;
during the retrieval it is just convolved to the instrument resolution applying $I_0$ weighted interpolation (Sect. A2). It is of
course possible to calculate it also directly at the instrument resolution, however this would mean its whole recalculation for
each calibration.

**A4   BrO absorption correction**

Within the OClO fit window used in this study also weak BrO absorption structures occur (Wahner et al., 1988). For the
retrieval without considering BrO, systematically enhanced OClO SCDs are retrieved for periods where the presence of OClO
is not expected. Since BrO has a strong yearly cycle with a maximum in winter (Sinnhuber et al., 2002), it is important to
eliminate possible interferences with the retrieved OClO SCDs as far as possible in order to not misinterpret the OClO data.

Within the chosen OClO fit window the BrO cross section shows only very weak spectral features. Thus including the BrO
cross-section directly in the OClO fit as a free fit parameter, leads to unreliable BrO results and accordingly in a systematic
negative offset in the retrieved OClO SCDs. To overcome this problem, we use BrO SCDs $S_{BrO}$, provided by Warnach et al.
(2019), retrieved in another fit window (330.6–352.75 nm) better suited for the BrO retrieval to subtract the BrO absorption
from the measured spectra before performing the OClO DOAS fit.

The correction term:

$$\tau_{BrO} = -R \cdot S_{BrO}\sigma_{BrO} \tag{A21}$$

is subtracted from the left term of Eq. (1). $\sigma_{BrO}$ is the BrO cross-section (Wahner et al., 1988) within the OClO fit range con-
volved with the ISRF. Since at high SZAs radiative transfer differences between different wavelengths can become important,
a scaling factor $R$ is used, defined as:

$$R = \frac{S_{380}}{S_{340}} \tag{A22}$$

where $S_{380}$ and $S_{340}$ is the BrO SCDs simulated by a radiative transfer model at the wavelengths of 380 and 340 nm,
respectively. For $R$ used in the retrieval, a BrO profile with peak at 17 km altitude and a Gaussian shape with FWHM of 6
km is assumed. $R$ of course is sensitive to these settings but it is better to apply this correction even with possibly inaccurate
settings than not to apply it (see the sensitivity studies in Appendix B6). Fig. A1 shows the dependency of $R$ as function of
SZA for 3 different BrO peak heights showing the importance of this correction at high SZAs (scaling factor is 1.6 at SZA of
90°). The uncertainty related to the peak altitude by 3 km is up to 0.3.





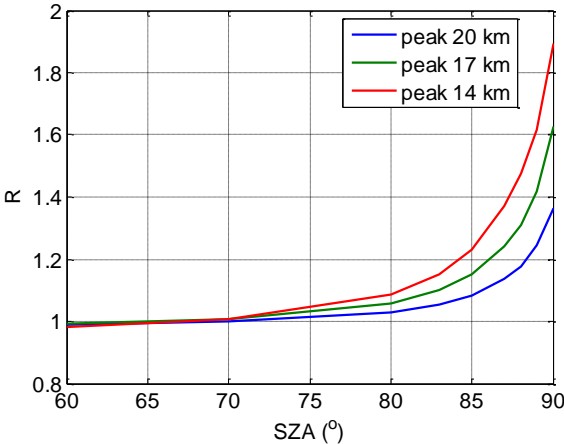

**Figure A1.** Scaling factors $R$ accounting for the BrO SCD differences at different wavelengths as function of the SZA. For the BrO correction in the OClO retrieval the factors modelled for a BrO profile with a peak at 17 km are used. Plotted are also values for 2 additional BrO profile peak altitudes.

## Appendix B: Fit sensitivity studies

We investigate the effect of different retrieval settings on the retrieved OClO SCDs by applying modifications with respect to the standard fit scenario described in Sect. 2, Table 1. The considered cases with the corresponding changes are given in Table B1. The retrieval performance for the same days (25 Aug 2018 for (NH) and 25 Dec 2018 for (SH), 25 Nov 2018 (NH) and 25 Apr 2019 (SH), as well as 25 Dec 2018 (NH) and 25 Aug 2019 for SH) as introduced in Sect. 2.2, Fig. 1, representative for different atmospheric conditions, are studied.

The obtained daily mean OClO SCDs for the defined cases are plotted in Fig. B1 for all 6 days. Since the standard deviations of the gridded means are very similar for the different months except 25 Dec in the SH (where the relative differences between the cases are still very similar as for the other days), an example for the standard deviation of the mean for just one of the days (25 Dec in the NH) is shown in Fig. B2. In addition, Fig. B3 displays the autocorrelation coefficients calculated as described in Sect. 2.2.2 as function of the across track lag for a zero along track lag. The figure allows a quantitative judgement about the magnitude of any systematic spatial artefact structures between the different retrieval settings. Figs. B4 and B6 provide the retrieved binned OClO SCDs and Figs. B5 and B7 their differences with respect to the standard scenario. These plots are shown for 25 Nov 2018 (NH) and also for 25 Dec 2018 (SH) in order to illustrate the seasonal differences. It is again found that the autocorrelation coefficients for 25 Dec 2018 (SH) are quite different from those of the other investigated days.

In the following we discuss the findings for the different cases





**Table B1.** Cases for the sensitivity study

| Nr. | Implemented modifications with respect to the standard fit settings |
|---|---|
| 1 | Calculation of the OClO SCD from fitted OClO + OClO$\times\lambda$ terms at 377 nm (instead of 379 nm) |
| 2 | Retrieval without the OClO$\times\lambda$ term |
| 3 | Slightly different fit window I (363 – 391 nm) (as e.g. by Kühl et al., 2004a) |
| 4 | Different fit window II (365 – 389 nm) (as e.g. by Oetjen et al., 2011) |
| 5 | Fraunhofer reference as daily mean of the earthshine spectra (Sect. A1.1) instead of the mean of normalized earthshine spectra (Sect. A1.2) |
| 6 | BrO correction (Sect. A4) taking the wavelength dependency of the BrO SCD into account assuming (Fig. A1) a profile peak at 20 km (instead of 17 km) |
| 7 | BrO correction (Sect. A4) without accounting for the wavelength dependency of the BrO AMF |
| 8 | BrO correction (Sect. A4) not applied |
| 9 | BrO correction not applied, but the BrO cross-section included as a fit parameter in the OClO fit |
| 10 | NO$_2$ Ring spectrum (Sect. A3) is excluded |
| 11 | Only Ring spectra for one temperature (280 K) |
| 12 | Fit without the slit function pseudo absorbers (Beirle et al., 2017) |
| 13 | Offset correction $\lambda^2/I0$ term excluded |
| 14 | Standard convolution for the trace gas cross sections applied instead of the intensity weighted (I$_0$) convolution (Sect. A2) |
| 15 | Same as 14 but with the offset correction $\lambda^2/I0$ term excluded |

## B1  Case 1: Wavelength assumption for the OClO$\times\lambda$ term

As suggested in Puķīte and Wagner (2016), when higher order terms are applied in the fit, the wavelength for the calculation of the trace gas SCD from the fitted coefficients should be selected empirically from within the fit interval because the agreement with the true SCD can vary with wavelength (as shown e.g. in the sensitivity studies for BrO in Puķīte et al. (2010)). To demonstrate the effect of the wavelength selection for the retrieval of the OClO SCDs, we changed the recalculation wavelength to 377 nm instead of our standard wavelength of 379 nm. The retrieved OClO SCDs are reduced by up to $0.5\times10^{13}$ cm$^{-2}$ at large SZAs (blue line in the left plots in Fig. B1) compared to the standard setting for the summer days. The shape of the SZA binned mean SCD dependence is, however, very similar to that of the standard scenario. The standard deviation is even slightly better (blue line in Fig. B2, left panel) which is expected because the wavelength is closer to the center of the fit interval. This case, however, has an increased autocorrelation (see blue lines in left plots in Fig. B3) being especially strong for 25 Dec 2018 (SH) with more than 2$\times$ higher values at the wings of the function. It shows an autocorrelation of 5 per mill even at a lag distance across track of 70 pixels while the value for the standard scenario is close to zero already for much shorter lags. Also for the 25 Nov 2018 the autocorrelation is much larger than for the other fit scenarios and is still above zero at large lags. The



**Figure B1.** Retrieved daily mean OClO SCD as function of SZA (resolution 0.2°) for the different cases of the sensitivity study described in Table B1 in comparison with the standard scenario for selected days in both hemispheres.

increase in the spatial structures can be clearly seen also in the maps of the binned OClO SCDs (Figs. B4 – B7, upper row, left center plot).





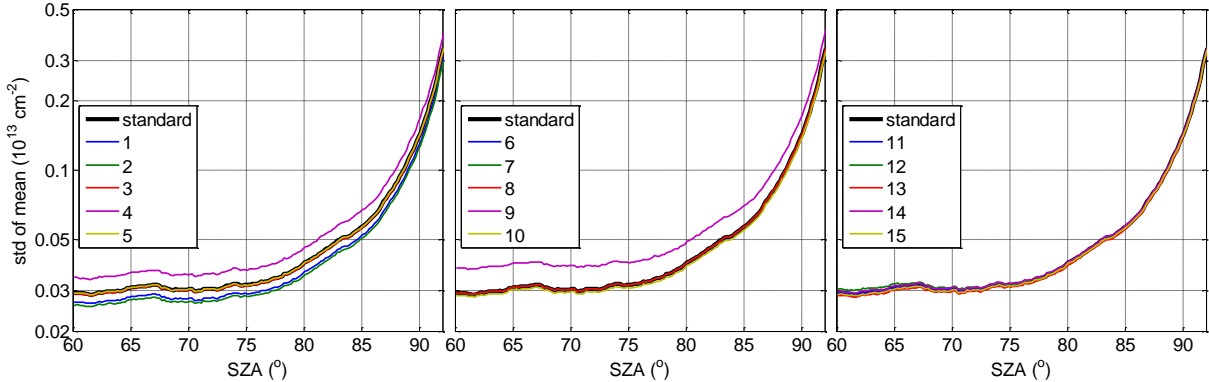

**Figure B2.** Standard deviation of the daily mean OClO SCD for 25 Dec 2018 in the NH for the different cases of the sensitivity study described in Table B1 in comparison with the standard scenario for selected days in both hemispheres.

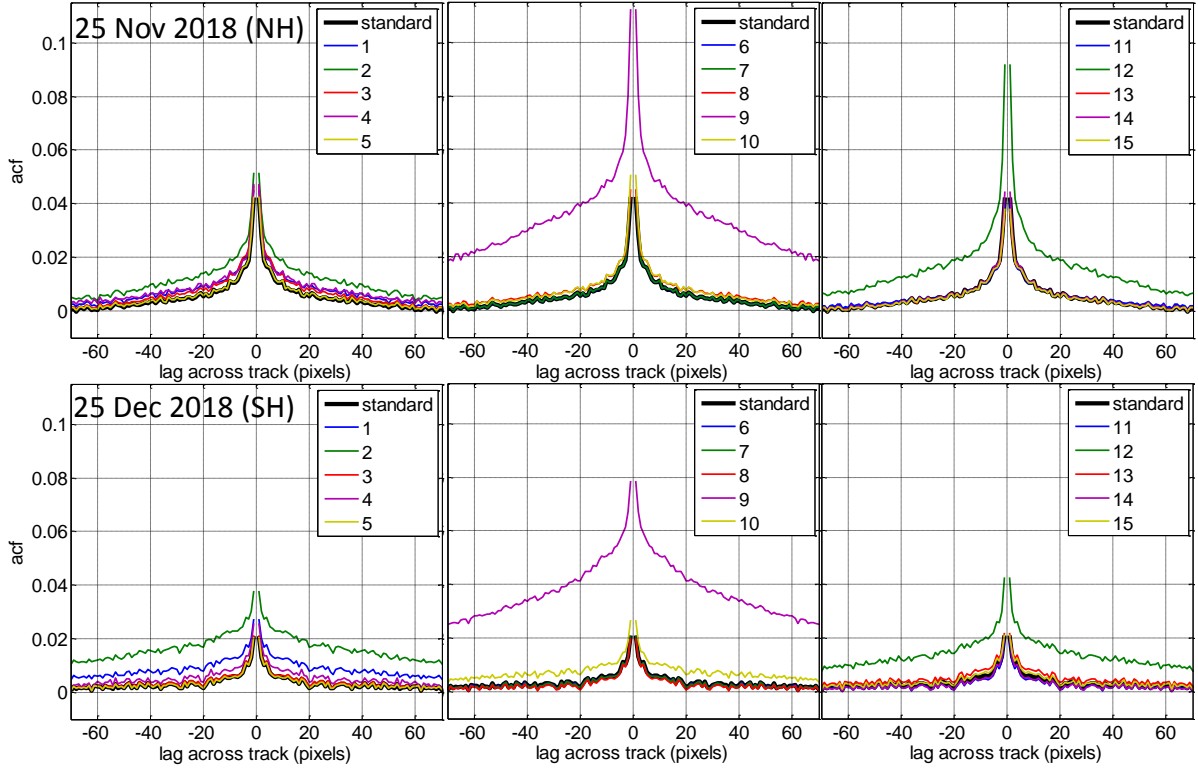

**Figure B3.** Autocorrelation coefficients calculated as in Fig. 2 but for different cases of the sensitivity study as function of lags across track with a lag along track of 0. Result for 25 Nov 2018 (NH) (top panel) and 25 Dec 2018 (SH) (bottom panel).



**Figure B4.** OClO SCDs for 25 Nov 2018 in the NH binned on a 0.2°×0.2° grid in an equidistant in latitude coordinate projection. Areas with cloud fractions (CF) below 5% are shadowed.



**Figure B5.** Differences between the OClO SCDs shown in Fig. B4 and those of the standard scenario.

This case shows that for the selection of the wavelength for the SCD calculation from the retrieved coefficients from the higher order terms, a trade-off has to be found between precision and accuracy.

**Figure B6.** Same as Fig. B4 but for 25 Dec 2018 in the SH. Note the different treatment with respect to the descending orbit parts (with higher SZAs): in the western hemisphere, only data for the ascending parts are shown, while in the eastern hemisphere data of the descending parts are plotted when available.





**Figure B7.** Same as Fig. B5 but for 25 Dec 2018 in the SH.





## B2   Case 2: Skipping OClO×λ term

In this case only the constant OClO cross-section term is fitted and the obtained SCDs are compared. In this case the retrieved SCDs are even (∼2×) lower than for case 1, being again largest at high SZAs (up to $1×10^{13}$ cm$^{-2}$, green line in the left plots in Fig. B1). The standard deviation (green line in Fig. B2 agrees quite well with that of case 1, indicating that adding higher order terms does not reduce the random retrieval error per se. The autocorrelation (green line in the left plots in Fig. B3) for this scenario is, however, more than 4× larger than for the standard scenario indicating the appearance of even larger systematic

spatial structures than for case 1. The binned OClO SCDs (Figs. B4 – B7, upper row, right centre plot) confirm the increase of the systematic structures.

Thus we can conclude here that it can be beneficial to add the higher order lambda term for the trace gas of interest to improve the performance of the retrieval with respect to systematic errors.

## B3   Case 3: Fit window 363–391 nm

The selection of the fit window used in our study (363–390.5 nm) is based on previous studies in our group (e.g. Kühl et al., 2004a). While on average the result for this scenario (red lines in the left plots in Figs. B1 and B2) agrees well with the result for the standard retrieval, the autocorrelation coefficients for this scenario (red line in left plots in Fig. B3) are somewhat larger in some cases like for 25 Nov 2018 (NH) where it is as large as for the case 1 for shorter lags. In the maps of binned OClO SCDs (Figs. B4 – B7, upper row, right plot) also small differences can be seen.

## B4   Case 4: Fit window 365–389 nm


We tested also the fit window (365 – 389 nm) which has been used by Oetjen et al. (2011). Unfortunately the use of this fit window gives much worse results which can be seen already in the plot with the gridded mean OClO SCDs as function of SZA as a systematic "wavy" structure (magenta line in the left plots in Fig. B1). Interestingly, this structure has its peaks and depths at the same SZA values for all seasons and both hemispheres, possibly indicating some instrumental problem (we see similar

structures in the results for different cases in general, only the amplitude is different). The interanual variation for this scenario is larger: in the autumn and winter months there is a large positive offset with respect to the standard scenario, while in summer this offset, still positive, is smaller. Interestingly, in summer the results of this scenario are closer to the expected value of zero than those of the standard analysis. This wavelength range selection, however, increases the random error (magenta line in Fig. B2. The autocorrelation is also increased (magenta line in the left plots in Fig. B3) compared with the standard scenario. But

since the autocorrelation coefficients are calculated at lower SZAs, the large OClO variability at high SZAs does not show up here. The binned OClO SCD maps (Figs. B4 – B7, middle upper row, left plot), however, shows these "wavy" structures like rings at high SZAs.





## B5 Case 5: Fraunhofer reference as daily mean of the earthshine spectra

In this setting Fraunhofer reference calculated as daily mean of the earthshine spectra (Appendix A1.1) instead of the mean of
the normalized earthshine spectra (Appendix A1.2) are used for the retrieval. The dark yellow lines in the left plots in Figs. B1,
B2 and B3 show perfect agreement with the standard scenario. The binned difference (second row, middle left plot in Fig. B5
shows only a very small detector striping.

## B6 Cases 6–8: BrO correction

Cases 6–8 demonstrate the effect of the application of different aspects of the BrO correction (Appendix A4). In case 6 just
the altitude of the profile for which the ratio of the BrO SCD between 380 and 340 nm (Fig. A1) are calculated is modified.
The peak of the profile is assumed to be at 20 km instead of 17 km. In case 7 the wavelength dependency is ignored (i.e. the
same SCD is assumed in both spectral ranges) while in case 8 BrO correction is not applied at all, i.e. the BrO absorption
contribution is not subtracted from the measurement spectra.

The blue, green and red lines, middle column in Fig. B1, respectively, show the systematic effects of these settings. For all
these cases a positive offset with respect to the standard scenario is found being larger for the days where more BrO is observed
(in autumn and winter). When the BrO correction is not applied, artificially enhanced OClO SCDs of $3$–$4\times10^{13}$ cm$^{-2}$ (25 Nov
in 2018 and 25 Apr 2019 in SH) at an SZA of 90° are retrieved. Applying the BrO correction but without considering the
SCD dependency on wavelength, the offset is corrected up to SZAs of ∼85° because the BrO SCDs are almost independent on
the profile shape and wavelength at short wavelengths. At a SZA of 90°, the retrieved artificially enhanced OClO SCDs are,
however, decreased by about a factor of 2 for the two autumn days. Thus, an offset at a level of ∼$2\times10^{13}$ cm$^{-2}$ remains. This
is further corrected by considering the BrO SCD wavelength dependency between the fit window of the BrO retrieval and the
OClO fit window. Depending whether the BrO profile at 17 km (as in the standard settings) or 20 km is chosen, the result varies
by less than $1\times10^{13}$ cm$^{-2}$. The effect of the BrO correction for the days in summer where low BrO levels are expected is much
smaller and is even leading (or contributing) to a negative offset of around $1\times10^{13}$ cm$^{-2}$. The BrO correction settings have
no effect on the random error (B2, middle plot, where the respective blue, green and red lines overlap with the standard case).
Also the autocorrelation coefficients (blue (overlaid by green), green and red (largely overlaid by yellow for the 25 Nov 2018)
lines in the middle plots in Fig. B3) match (except for case 8 in autumn) the result for the standard case as the implemented
settings have an effect for high SZAs (cases 6 and 7) or for large BrO SCDs (case 8 in the plot for 25 Nov 2018 in the NH). The
binned OClO SCD maps (Figs. B4 – B7, middle upper row, the two plots on right side for cases 6 and 7, as well as the middle
lower row on the left side) also indicates mostly latitude (i.e. SZA) driven differences with respect to the standard scenario.
Some additional variation also in East - West direction for case 8 (middle lower row, left plot in the same figures) caused by a
larger stratospheric dynamics in BrO probably explains the slightly increased autocorrelation for the autumn day.





### B7   Cases 9: BrO included as a fit parameter

It would be a more straightforward approach to use a BrO cross-section directly in the OClO retrieval as additional fit parameter
instead of the BrO correction described above. However, the results of this sub-section illustrate that this is not a good choice:
The retrieved OClO SCDs suffer from a strong offset exceeding the systematic uncertainties even at relatively low SZAs (Fig.
B1, magenta line in the middle column), and it also shows the "wavy" structures besides as also seen in case 4. Also the random
error is significantly larger than for most other settings (magenta line in middle plot in B2). The large systematic structures
show also strong spatial variation as indicated by the results of the autocorrelation analysis (magenta line in the middle plots
in Fig. B3) and in the binned OClO SCD maps (Figs. B4 – B7, plots in the middle lower row, middle left).

### B8   Case 10: Skipping $NO_2$ Ring absorption spectrum

In this case the $NO_2$ Ring absorption spectrum (Sect. A3) is excluded from the retrieval. As a consequence, a negative offset of
up to around $2 \times 10^{13}$ cm$^{-2}$ with respect to the standard scenario is observed towards larger SZAs (Fig. B1, yellow line in the
middle column). Also in the binned OClO SCD maps (Figs. B4 – B7, plot right from and below the middle) a clear latitudinal
gradient can be observed with respect to the standard scenario. As expected, this deviation is larger for the summer months
where more stratospheric $NO_2$ appears at higher latitudes than in the winter months. The random error is slightly smaller than
for the standard scenario (dark yelow line in the middle plot in B2). The autocorrelation coefficients (dark yellow line in the
middle plots in Fig. B3) are increased and are also larger in summer obviously corresponding to the increased impact of the
$NO_2$ spatial distribution on the OClO retrieval.

### B9   Case 11: Ring spectra at just one temperature (280 K)

The consideration of Ring spectra in the retrieval is one of the parameters we found, to which the retrieval settings are sensitive
to. Excluding the second Ring spectra for the temperature of 210 K from the retrieval (blue lines in the right plots in Fig. B1)
also introduces a seasonally dependent offset (in this case being positive at larger SZAs) with respect to the standard scenario.
Opposite to the results in case 10 (skipping the $NO_2$ Ring spectrum), in this case the offset is larger for days in autumn and
winter. There is practically no effect of this change on the random error (B2, right plot, blue line). Interestingly, also the
autocorrelation coefficients (Fig. B3, right plots, blue line) are a bit larger than for the standard case. A potential explanation
is that the spatial structures have a much coarser structure which is not well covered by the single orbit subsets used in the
autocorrelation analysis Also the binned OClO SCD maps (Figs. B4 – B7, right plot, lower middle row) show differences in
the systematic structures which besides the latitudinal variation vary also along longitude.

### B10   Case 12: Retrieval without slit function pseudo absorbers

The parameterisation of slit function changes (Beirle et al., 2017) is also important for our retrieval: Already in the SZA re-
solved daily mean plots (green lines in the right plots in Fig. B1) a more pronounced fluctuation is found if this parameterization
is not considered. The autocorrelation analysis (green lines in the right plots in Fig. B3) shows both much higher correlations





for short as well as for long lags. This increase could be caused by the slit function variability, e.g. by the inhomogeneous
illumination of the slit, the polarization sensitivity or other factors. The much larger spatial variation can be seen also in the
binned OClO SCD maps (Figs. B4 – B7, left plots in the bottom row). The setting of this case, however, has no effect on the
random error (B2, right plot, green line).

### B11   Case 13: Offset correction $\lambda^2/I_0$ term excluded

In our standard retrieval we consider an intensity offset correction up the second order (i.e. the $1/I_0$, $\lambda/I_0$ and $\lambda^2/I_0$ terms)
There is a very small effect when the $\lambda^2/I_0$ term is skipped (red lines in the right plots in Fig. B1 – B3, and the middle left
plots in the bottom row of Figs. B4 – B7). It should, however, also be noted that a larger discrepancy between the results of this
case and the standard scenario is found in the autocorrelation coefficients for the 25 Dec 2018 (SH). The fact that these orbital
subsets are observed over Antarctic leads to larger measured radiances and thus might lead to a different straylight contribution.

### B12   Case 14: Application of a standard convolution

In this case study we apply a standard convolution for the trace gas cross sections instead of the intensity weighted ($I_0$)
convolution described in (Sect. A2). As observed in Fig. B1 (magenta lines, right plots) and the binned OClO SCD maps
(middle right plots at the bottom row of Figs. B4 – B7), this setting introduces a small but still distinguishable positive offset
with respect to the standard scenario. This offset is larger in summer, indicating that it might mainly be caused by differences
in the $NO_2$ absorption cross-section treatment whose absorption is larger in this season.

### B13   Case 15: Application of a standard convolution and an offset correction $\lambda^2/I_0$ term excluded

During the sensitivity studies described above, we found that in some cases there can be a larger effect with respect to the offset
correction $\lambda^2/I_0$ term if the standard convolution is performed. Indeed, Fig. B1 (dark yellow lines, right plots) shows a better
agreement of this case (when both standard convolution for trace gas cross-sections is used and the $\lambda^2/I_0$ is omitted) with the
results of the standard scenario than the case 14. The results of case 15, however, have a higher autocorrelation (Fig. B3, right
plots, dark yellow line) than those of the standard case.

### Appendix C:   Comparison with Kiruna zenith sky measurements with Ring spectra at only one temperature and without the OClO×$\lambda$ term

In this Appendix we repeat the comparison between the zenith sky measurements at Kiruna and TROPOMI as in Sect. 3.1 but
with the Kiruna OClO SCDs retrieved using just 2 Ring spectra (scaled and not scaled with $\lambda^4$) at only one temperature (250
K) and without the OClO×$\lambda$ term in the DOAS fit.

The timeline comparing both data sets and their difference are shown in Figs. C1 and C2 in the same manner as it is done
in Figs. 4 and 5 (Sect. 3.1), respectively. Along somehow larger absolute discrepancies (up to $8\times10^{13}$ cm$^{-2}$ for SZA at and
below 90°) than in Sect. 3.1, also pronounced inconsistencies are observed in a form of an offset between the TROPOMI





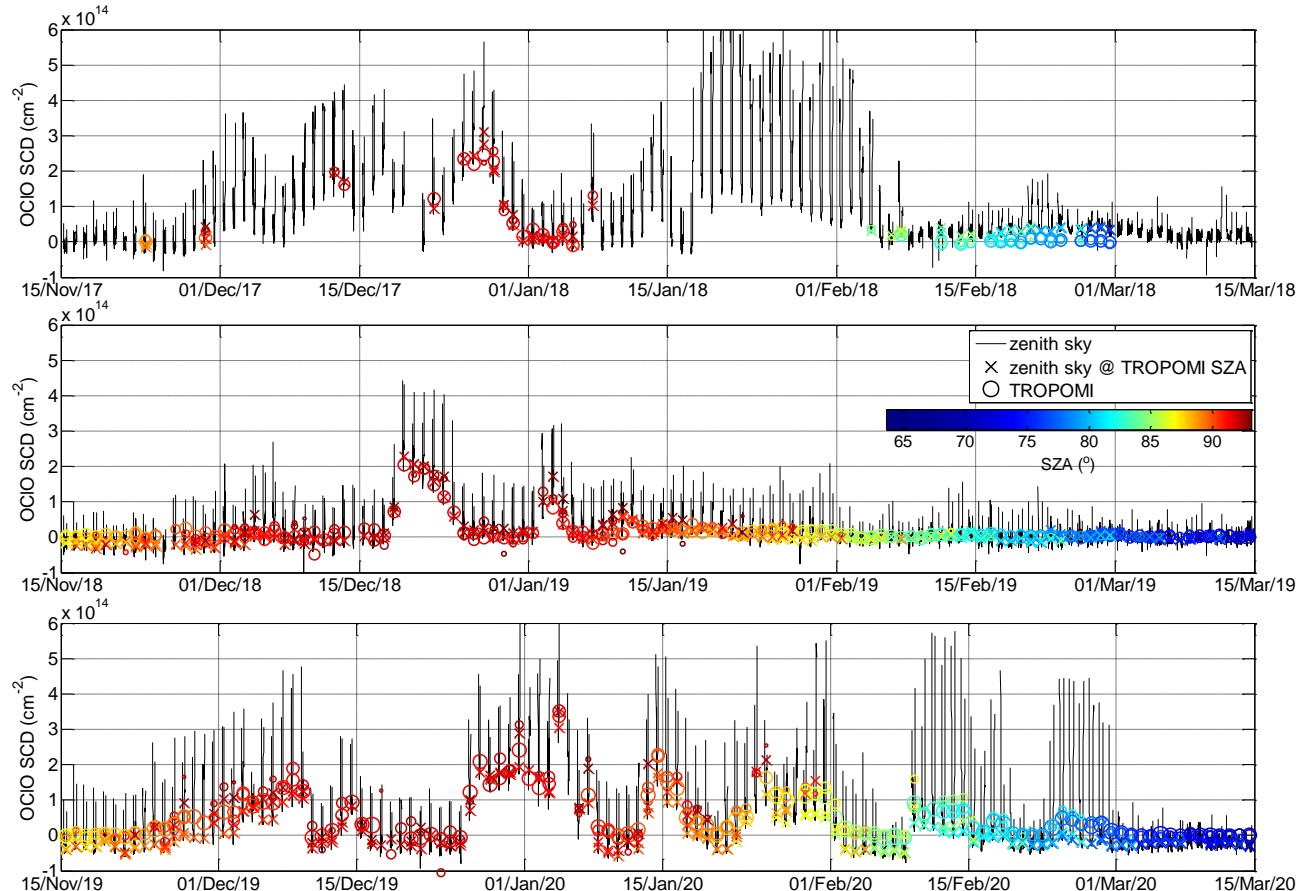

**Figure C1.** Same as Fig. 4 but comparison with ground based OClO SCDs at Kiruna retrieved with Ring spectra at a single temperature and without the OClO$\times\lambda$ term in the DOAS fit.

.

and Kiruna SCDs being different for different winters. The difference between the datasets of both instruments also varies
on weekly or semi-monthly basis inducing a 'wavy' pattern in the difference plots. Also an overall trend can be recognized
with an increasing difference increasing from the beginning of winter towards spring. Fig. C3 illustrates the worsening of
the comparison with respect to the agreement found in Sect. 3.1 (Fig. 6) even more clearly: A larger scatter in the difference
between the collocated TROPOMI and zenith sky DOAS measurements as function of the SZA (x-axis) is visible in the left
plot. The seasonal variability in the offset shows a clear dependency on SZA being at maximum as high as $\sim 3\times10^{13}$ cm$^{-2}$
(i.e. up to around $3\times$ larger than for the settings in Sect. 3.1) for SZAs between $85°$ and $90°$. Also the standard deviation of the
differences is up to $\sim 3\times$ larger (for SZAs below and around $90°$). The scatter plot (Figure C3, right) between the TROPOMI
and zenith sky data shows a worse correlation. The slope for the orthogonal regression is now only 0.75 (0.94 in Sect. 3.1)
whereas the offset is $9.4\times10^{12}$ cm$^{-2}$ ($1\times10^{13}$ cm$^{-2}$). The correlation coefficient between both datasets is 0.86 (0.94).

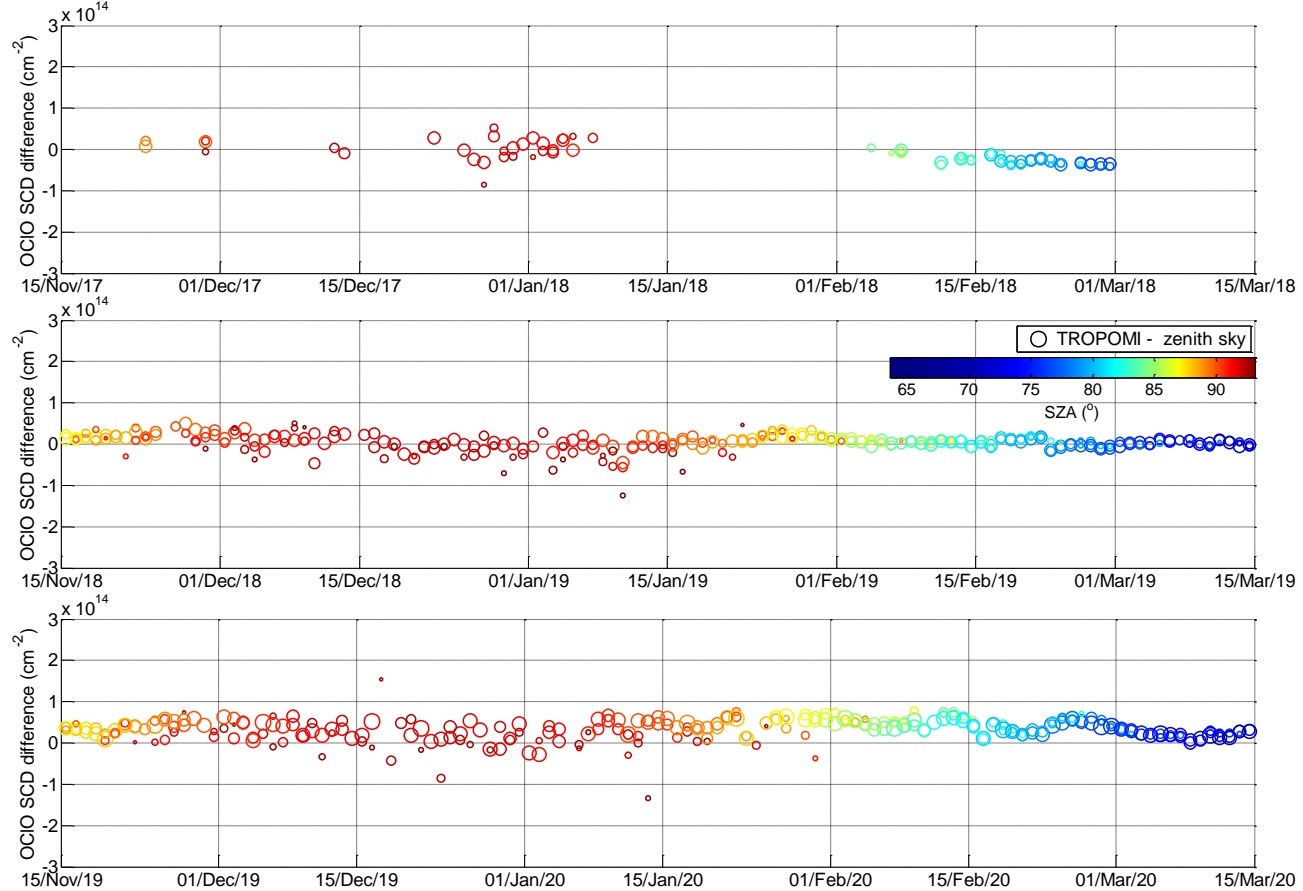

**Figure C2.** Same as Fig. 5 but considering ground based OClO SCDs at Kiruna retrieved with Ring spectra at a single temperature and without the OClO×λ term in the DOAS fit for the calculation of the difference.

.

*Author contributions.* J.P. with support of C.B. S.D. M.G. and T.W. performed the study and analysed the results. C.B. with support of J.P. and T.W. retrieved OClO SCDs from TROPOMI measurements. M.G. with support of J.P. and T.W. retrieved OClO SCDs from zenith-sky measurements at Kiruna. U.R and C-F.E. maintained the zenith-sky instrument at Kiruna. U.F. provided OClO SCDs from zenith-sky measurements at Neumayer. A.M. and A.R. provided the preliminary S5p+I OClO SCDs. J.P. prepared the manuscript with supervision by T.W and comments by all co-authors.

*Competing interests.* No competing interests are present





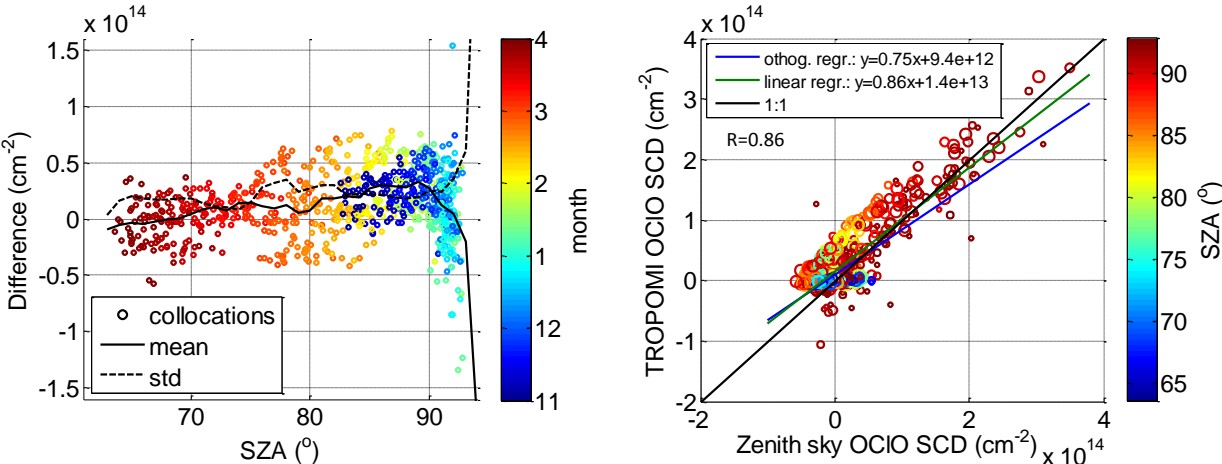

**Figure C3.** Same as Fig. 6 but comparison with ground based OClO SCDs at Kiruna retrieved with Ring spectra at a single temperature and without the OClO×λ term in the DOAS fit.

*Acknowledgements.* We acknowledge ESA and SP5 team for providing TROPOMI l1b data. We thank Simon Warnach for providing the TROPOMI BrO dataset.



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
