# Peer review of "Retrieval algorithm for OCIO from TROPOMI by Differential Optical Absorption Spectroscopy"

_Atmospheric Measurement Techniques, 2021_

## Author Comment (AC1)

Response to the Reviewer #1

We thank the Reviewer for the constructive review and address the comments below.

*The authors present a new algorithm to retrieve OClO from measurements of the TROPOMI satellite instrument. Overall, the paper presents valuable information and recommended DOAS settings for OClO retrievals from space that deserves to be archived. I believe the paper should be published in AMT. The authors present convincing comparison between TROPOMI and ground-based OClO SCDs, which clearly demonstrate the value of the satellite dataset. The authors present a number of tests and new settings. Some of these are new and make a lot of sense. However, the impression is also that many complication is introduced that is not really necessary. What is missing is a summary of the retrieval settings and choices that has the largest impact on the accuracy. The English is not very good and could be improved.*

We completely agree with the reviewer that some of the settings do not give large improvement as already shown in the sensitivity studies (Appendix B) and thus are optional. We will provide the requested summary in the main text of the manuscript as a separate subsection (Sect. 2.2.4):

"Sect. 2.2.4.Sensitivity to retrieval settings

[revised manuscript text omitted]

We tried to improve the English, but also rely on the proofreading service provided by Copernicus office with respect to the English of the manuscript.

*-Retrieval settings: Section 2.1*

*-Calibration: It is unclear whether it is performed on solar measurements or on mean radiance spectra. In the latter case, atmospheric SCDs should be fitted as part of the calibration step?*

The calibration is done on the earthshine reference spectra. Absorption by ozone and NO2 are considered here. We add this information to the algorithm description.

*The description in Appending A1.1-2 is unclear. I don't understand what is tau_i and I_i and what the proposed weighting (section A1.2) is supposed to solve.*

tau_i is the signal of the absorption features to be fitted by the retrieval coming from a single pixel, I_i is the intensity of the signal, i.e. for pixels above the clouds there will be a stronger signal than for clear scenes.

The weighting approach ensures that the mean offset of the fit result in the logarithmic domain (optical depth space) in the offset region is zero because the absorption signals are not weighted anymore by the intensities of different pixels. We show in sensitivity studies that this (theoretically better) setting in the OClO retrieval does not play a role which is a good sign: it means that there are no (or at least no spatially variable) OClO signal in the reference region which could provide an offset in case of applying the mean of the spectra in the reference region. Also it indicates that there are no other (no spatially variable) unexplained spectral structures highly correlating with the absorption structures of OClO that also would provide an offset in the retrieved OClO SCDs.

We added the required explanation "…tau_i being the signal of absorption features from a single pixel i to be contributing to absorption parameters fitted by the retrieval and the intensity I_i being the intensity of the signal, e.g. for pixels above the clouds there will be a stronger signal than for clear scenes."

Also the motivation in the last paragraph of Sect A.1.1. was improved (as a consequence of Eq. A2) "Thus, such a reference spectrum in the DOAS analysis generally would lead to an offset for the fitted parameters even in the reference region if their SCDs are not homogeneous in this region. Also if there is no expected absorption of a particular absorber in the reference region (like it is the case for OClO), the potential errors and the incompleteness of the representation of the atmospheric state by the DOAS model can in theory induce an offset because part of the signal could interfere with the absorption cross sections of the considered absorbers. Fortunatelly the performed sensitivity studies (Appendix B5) show no additional effect from these considerations to the retrieved OClO SCDs. Nevertheless, we eliminate even theoretically such an offset by considerations in the next subsection."

In Sect A.1.2. we added a note that the calculation of the weighted mean reference requires practically no additional calculation effort.

*-The sentence on l106 "The effect for this application is however negligible" is strange. Why introducing something in the text which has no effect?*

It is introduced because from the theoretical point of view it is a better setting avoiding an extra offset (see also previous point) and it is always better to use theoretically better approaches if they do not require any additional effort.

We reformulated the two sentences referred here to better motivate the use: "The use of the normalized spectra (Appendix A1.1.2) for the calculation of the daily mean (at practically no additional calculation effort) ensures that also spectral features that are not related to OClO but correlate with its cross-section are not producing an artificial offset. The effect of this theoretically better approach for this application is however negligible."

*-the description of the Ring effect is unclear. It is explained that Ring spectra are calculated at 2 Temperatures from the reference spectrum. Do you mean the reference for SZA 60-65°? If yes, I don't understand how it is calculated. Are the 4 Ring cross-sections fitted?*

The Ring spectra are calculated from the mean earthshine reference, as it would be calculated from a Sun spectrum. The Ring spectra calculation from the earthshine reference corresponds to the earlier research referenced in the study (e.g. Kühl et al., 2004b, 2006, 2008) Yes, all 4 Ring cross-sections are included in the fit.

We have also performed a test using measured Sun irradiance spectra for the calculation of the Ring spectra but did not find an improvement. Also the use of the Ring spectrum as defined in the S5P+I product does not provide an improvement. For illustration see Fig. R1.

We also made this more clear in the text: "The Ring effect is accounted for by Ring spectra calculated at two temperatures (280K and 210K) in order to account for dependency of Ring structures on temperature, which we found is important (see Appendix B9). The two Ring spectra are calculated from the Earth-shine reference spectrum and included in the fit. Each also is scaled with $\lambda^4$ according to Wagner et al. 2009 (additional two spectra). The use of an Earth-shine reference spectrum for the calculation of the Ring spectrum is in accordance with previous studies (e.g. Kühl et al., 2004b, 2006, 2008) and is found to give a slight improvement with respect to the calculation of the Ring spectra from measured Sun irradiance spectra."

A statement with respect to Ring spectrum as for S5P+I is also added at the end of Sect. 4. See the response to the Reviewer 2.

[Figure]

Fig. R1. Top: Retrieved daily mean OClO SCD as function of SZA (resolution 0.2°) (similar as Fig. B1 in the manuscript) for days in two different seasons for the retrieval using a Sun reference spectrum for the calculation of the Ring spectra (21), using the same Ring spectrum as for the S5P+I retrieval (25), using the Ring spectrum as for the S5P+I retrieval and also scaling it by $\lambda^4$ (26) in comparison to the standard settings (standard). Bottom: calculated autocorrelation coefficients for the mentioned cases (similarly as in Fig. B3 in the manuscript).

*-Section 2.2*

*To help the reader, I suggest to add directly in Fig1 the indication of which days are expected to have enhanced OClO or not. In Fig1 left, the marker "x10" should be "divided by 10"?*

Many thanks for these suggestions, which we implement as suggested.

*-Figure 3: it would make sense to show the standard error also or instead of the std (which is already shown in Fig1).*

We do not understand this suggestion. Fig. 3 already shows standard deviation of the binned mean (which assuming independent random variables is standard error of the mean). Fig. 1, however, shows mean standard deviations of single pixels. In other words, Fig. 3 does not show the same quantity as Fig. 1 and already illustrates the standard error of the binned data.

**-Section 3:**

**-**the ground-based data are not analyzed using the same settings as used for TROPOMI. It is not fully clear to what extend and how this can explain the observed differences. E.g. in Fig6 left, there is a clear offset of ~1e13 cm-2 between TROPOMI and ground-based data for low SZA. Is this related to different DOAS settings, sampling bias, other? Please discuss this in the text.

The comparison was performed with datasets of ground base datasets with settings obtained by independent studies. We already demonstrate the effect of a different setting for Kiruna (Appendix C) showing a worse result, nevertheless also there the offset is still below 2e13 cm-2. For Kiruna we have found that the usage of a reference spectrum from a different day can slightly modify the offset. Nevertheless the offset is below the accuracy of the retrieval and thus can be neglected which we will add as a statement to the manuscript. A more detailed investigation, in particular a study towards unifying OClO retrieval settings for different instruments, would be an important investigation on itself and as such is more as just a validation exercise and would reach far outside of the scope of this paper.

-Interestingly at Neumayer, the scatter of the SCD differences is much higher than for Kiruna. Is it because the SCDs range is larger? Or is there an instrumental related difference? Or something else? Please elaborate.

We can only speculate for the reason of larger scatter of the differences at Neumayer. Surely the SCD range and diurnal variation of the SZA is larger there because of the different latitudes of both sides and the specific TROPOMI orbital properties. There are also systematic differences in the difference plots from year to year (Fig. 8) which results in larger scatter as can be seen in Fig. 9, left. Also different retrieval settings compared with the Kiruna analysis could play a role but this cannot be confirmed without additional investigations.

We add: "We can speculate that the scatter for Neumayer in comparison with Kiruna is larger because of the different latitudes of both sites and the specific TROPOMI orbital properties along with the different retrieval settings."

**-Section 4:** it would be good to understand what is the dominating factor explaining the offset between the 2 OClO data sets. I imagine it is probably related to the use of irradiance as reference spectrum and it is likely the largest source of error of the retrievals.

We have found that the use of an irradiance spectrum as reference spectrum does likely not explain the differences as it would lead to a rather constant offset along the whole orbits. The comparison of both datasets to the ground based data revealed that the difference is limited to high SZAs for cases with low OClO. We added the information to the paper that the reference

spectrum can likely not explain the offset. See also the response to the Reviewer 2 for more details.

*-The Appendix B is hard to digest. I suggest to add a summary table (extending Table B1) in the main text with typical errors on the SCDs coming from the main sensitivity tests so that the reader can have a rapid idea of what matters and what not.*

This comment comes back to the general comment at the beginning of the review. We will follow the suggestion and add the suggested table and to the main text (see the answer to the comment about the summary of the sensitivity studies above).

*How the errors from the sensitivity studies are relevant compared to the typical OClO values and the differences from the validation exercise?*

Given that the differences in the validation exercise can include additional errors (e.g. different radiative transfer, collocation, instrumental and retrieval settings), it is probable that some of the retrieval settings for TROPOMI could be still relevant even if they cause smaller differences than found in the validation exercise. Thus we think that the errors from the sensitivity study should be put in a relation to the retrieval errors of the standard setting as already estimated in Sect.2.2.1. Thus we list in the summary table of the sensitivity study the errors of the standard scenario as well and compare the performance of the sensitivity cases with the performance of the standard scenario.

*-Sensitivity studies 5, 6, 13,14,15 have very little impact on the results. Consequently, one could argue that the related settings introduced are not really necessary. E.g. the mean of normalized earthshine spectra, the offset correction quadratic term could be optional.*

Yes, of course. We will add a discussion for this along with the information in the new table as suggested above.

***Minor comments***

*-Abstract: the first 10 lines are too generic for an abstract and should belong to the introduction section. "OClO" is defined twice in the abstract.*

We remove the sentences from the abstract. The information is already provided in the introduction.

*- "so called" -> "so-called"*

corrected

*-wording such as "Last but not least" should be avoided.*

We reformulate affected sentences

*-lines 315-316: a reference to a next section (Sect. ??) is erroneously made. Please remove.*

Here should be a reference to the manuscript https://acp.copernicus.org/preprints/acp-2021-600/ which in an earlier stage was part of the manuscript presented here. In that earlier version it was presented as a separate section to which a reference was made. After suggestion by the former editor, we split the original paper into a technical part (AMT) and a part with TROPOMI results and their meteorological interpretation (ACP). We oversaw the old formulation while splitting. We add now the correct reference to the second manuscript.

---

## Author Comment (AC2)

We thank the Reviewer for the constructive review and address the comments below.

*General Comments:*

*In this work, a new retrieval algorithm of the slant column densities (SCDs) of OClO is proposed. This algorithm, aimed to be applied to TROPOMI DOAS measurements, takes into account different spectral effects not considered in previous retrievals. A corresponding error analysis of some retrieval settings has been performed. The authors also present a comparison between OClO SCDs, obtained by TROPOMI trough the new algorithm, and from ground-based zenith DOAS measurements at Kiruna and Neumayer stations. The results show a very good agreement with these instruments (especially at Kiruna). The SCDs of OClO obtained in this work have been also compared to preliminary S5p+I OClO products during different periods of the year, showing similar SCD evolution but presenting an offset between both datasets.*

*All the manuscript (text and figures) is clearly presented. The new concepts and settings introduced in this new algorithm are exhaustively explained (appendixes), as well as the corresponding error analysis and sensitivity studies. I think that the results exposed in this work will be useful for the treatment and analysis of the OClO SCDs obtained by TROPOMI. Thus, I think the the paper should be publish in AMT. However, I think that some questions should be better clarified.*

*My main concern is that the authors of this work claim that the new algorithm improves the retrieval results, but looking at the comparison between SP5+I and the results of this work, can that be really stated? The new algorithm takes into account several fine effects that, in principle, should improve the OClO SCDs retrieval and decrease the corresponding errors. But, in practise, how can we say that the results of these work are better than those of SP5+I? For high SCDs, results are very similar, and for low SCDs the offset between both datasets cannot be explained. It is true that, as the authors explained, OClO observation is not expected when the temperatures are still warm, as it is observed in the results of this study (Figure 10). Contrarily, SP5+I results show a background level of OClO. But, it can be affirmed that the results of this work are better than those provide by the SP5+I? Did the authors compared the results of both algorithms with independent measurements (as those of Kiruna or Neumayer)? Please, explain better.*

In our opinion there is a misunderstanding here: We do not claim that our algorithm improves the retrieval results only looking at the comparison between our results and the preliminary SP5+I data. To claim that one algorithm is better (or not), not only the results themselves but also the errors should be considered. Since SP5+I is in development and a complete error analysis is not available yet (published), we cannot judge on this. Also given that our algorithm is developed independently of the SP5+I project, we just present the data of the comparison and state in conclusions that "A nearly perfect correlation (correlation coefficient being practically unity) is obtained with the comparison to the preliminary data of the operational S5P+I retrieval algorithm. In the S5P+I data however a systematic positive offset is found." Besides the statement that OClO observation is not expected when the temperatures

are still warm, we also do not state that our results are better than those of SP5+I and we do not want explicitly to pretend to give such a statement. We will now add "with respect to the presented algorithm" before "is found" of the cited sentence. The better agreement at high SCDs can very likely be explained by the different air mass factors leading to the regression slope being different from 1:1 and considering the regression offset. To better explain this we will add to the text in Sect. 4, L320: "The slope different from unity and the offset of the regression thus explain the good agreement at high OClO SCDs and the offset at low SCD values". To answer the last question, we have made a comparison of the data of both TROPOMI algorithms to the ground based measurements (see below) which however we think is not necessary to be shown in the paper as this would distract reader from the main focus of the paper on the current retrieval.

We found that the differences in the agreement of the TROPOMI datasets with the ground based data are limited to high SZAs for time periods with low OClO. Fig. R1 shows that the discrepancies for the S5P+I data are larger than those of our algorithm in comparison with the Kiruna data. The reason for this finding is that generally lower OClO SCDs are observed in Kiruna than for Neumayer where, due to many days with very high OClO SCDs, a much better overall agreement is achieved.

[Figure]

Fig. R1. The same as Fig. 9 (in the manuscript), with differences between TROPOMI OClO SCDs of our algorithm (left plots) or the preliminary S5P+I data (right plots) with zenith sky OClO SCDs measured at Kiruna (top) and Neumayer (bottom).

We will add/modify the statement about the discrepancies in the manuscript in the last paragraph of Sect. 4: "A comparison of S5p+I OClO SCDs with the ground based data (not shown here), performed in the same manner as the comparison in Sect. 3 between this study and the ground based data, showed generally a very similar agreement between S5p+I and the

ground based data. Larger differences between the S5p+I OClO SCDs and the ground based data than for our analysis was found for observations at high SZAs with low OClO SCDs, thus consistent with the findings in this section. As a consequence, the application of the solar irradiance instead of the earthshine spectrum as Fraunhofer reference cannot likely explain the differences because it would provide a similar offset for all SZAs. Also the use of a Ring spectrum as defined in the S5p+I preliminary product (not shown here) did not provide a better result. Thus we can speculate that the differences could be related to the usage of different fit windows, together with still uncompensated higher order effects in the current version of the S5p+I OClO fit as the consideration of the wavelength dependency of fit parameters becomes more challenging in larger fit windows. The differences might also be related to the implementation of the empirical terms in the S5p+I retrieval or instrumental effects, but such detailed investigations are beyond the scope of this study."

*Specific Comments:*

*I would like also to clarify some questions:*

- *Figure 1: Some days of different periods of the year for both, NH and SH, have been presented. Are those days representative of the corresponding periods? If it is the case, do you think that the bias introduced by SZA at each season could be, at least partially, corrected somehow?*

Yes, the days are representative for the corresponding periods. We make this more clear in the manuscript modifying the sentence on L145 "The days are selected to represent different atmospheric conditions" by adding "…at different time periods". So far we have not found a possibility to correct for the offset otherwise we would have done this already. Nevertheless the systematic error is low also in relation to other error components as shown later in the manuscript (e.g. at the end of Sect. 2.2.3). And as shown in the sensitivity studies the offset varies in a similar range depending on various fit parameters.

- *Page 11, line 225: why for clear sky cases the signal to noise would be lower?*

Typically clear sky cases have lower albedo, hence the backscattered or reflected light by clouds or Earth surface is lower. We add in brackets "due to a typically lower effective albedo"

*Technical Corrections:*

- *Page 4, line 102: "Earthshine" instead "Earth-shine", for coherence.*

  Corrected as suggested

- *Page 6, lines 123-124: A wavelength l=379 nm is selected for evaluation. Please, explain briefly why.*

  We add that "because this wavelength provides a good trade-off between precision and accuracy (see also the sensitivity studies in Appendix B1 and B2)"

- *Legend of Figure 1: ".. indicated in the legend on the right.", instead "left plot".*

  Corrected as suggested

- *Page 7, line 161: "In an ideal case,.."*

  Corrected as suggested

- *Page 17, line 316: "(Sect. ??)"*

  Here should be a reference to the manuscript https://acp.copernicus.org/preprints/acp-2021-600/ which in an earlier stage of the manuscript here was presented as a separate section to which a reference here was made. After suggestion by the former editor, we split the original paper into a technical part (AMT) and a part with TROPOMI results and their meteorological interpretation (ACP). We oversaw the old formulation while splitting. We add now the correct reference to the second manuscript.

- *Page 25, line 476: "depends on both", instead "depends both on".*

  Corrected as suggested

- *Page 25, line 489: "Within the chosen OClO fit window,..".*

  Corrected as suggested

- *Page 25, line 490: "cross section" instead "cross-section", for coherence.*

  Corrected as suggested throughout the paper

---

## Author Comment (AC3)

We thank the Reviewer for the constructive review and address the comments below.

*This manuscript presents a retrieval algorithm for OClO slant columns from TROPOMI measurements using the DOAS technique. To improve the accuracy of the retrieved data, the authors introduce additional fit parameters accounting for spectral effects which have previously not been accounted for and they provide a discussion of the uncertainty estimates including a novel application of an autocorrection analysis.*

*The authors show that their retrieval of TROPOMI OClO slant columns is in good agreement with ground-based zenith sky measurements made at two polar stations. They also compare their TROPOMI product with preliminary data retrieved with the operational TROPOMI OClO retrieval algorithm and discuss the observed differences.*

*The study is clearly presented in the manuscript, and in addition, the authors also provide substantial material describing relevant retrieval concepts and settings (Appendix A) and an extensive sensitivity study investigating the effect of the different retrieval settings on the OClO slant column data in comparison to a standard scenario (Appendix B). The paper is recommended for publication in AMT.*

*General comments:*

*While some of the aspects included in the uncertainty analysis of the retrieval include a novel approach (the application of autocorrelation for the systematic error analysis), stating that this is overall a new retrieval algorithm seems to me somewhat exaggerated since my understanding based on the manuscript is that the difference to existing algorithms is mainly that additional fit parameters have been used. If that is not correct, and the algorithm is indeed novel then please describe this clearer in the text.*

Of course the algorithm is still a DOAS algorithm (as reflected in the title) and the concept of DOAS limits the innovation just to using different fit parameters - can there be a novel DOAS algorithm then at all? The algorithm is a new DOAS algorithm for OClO from TROPOMI, also new is that additional fit parameters have been introduced for a DOAS retrieval for the first time. Of course we would not like to pretend to exaggerate and thus agree to replace "novel" in this context in the first sentence in conclusions with "new". We also modify the sentence in the abstract "Here we present a new retrieval algorithm of the slant column densities (SCDs) of chlorine dioxide (OClO) by DOAS" by adding "… from measurements performed by the TROPOspheric Monitoring Instrument (TROPOMI) instrument on board of Sentinel-5P satellite."

*Also, in the conclusions, the authors state that 'the detection limit is similar to the detection limits of earlier instruments' – i.e. that this has not really improved – but then, also in the conclusions, they state that 'Including these terms improves the retrieval results especially for low OClO SCDs'. Aren't these 2 statements contradicting each other?*

We want to say that the detection limit is similar to the detection limits of earlier instruments if we grid the measurements to 20x20 km2 area which is much smaller than the resolution of previous satellite instruments. Thus TROPOMI measurements provide a clear improvement with respect to previous instruments. Including the additional terms improves the retrieval results especially for low OClO SCDs. This statement is not contradicting as the inclusion of the additional terms improves the error budget and in particular the accuracy of the retrieval.

We modified the mentioned statements in the conclusions to make this more clear:

"Including these terms improves the accuracy of the retrieval results especially for low OClO SCDs."

and:

"Thus a detection limit of about 0.5-1 $\times10^{14}$ cm$^{-2}$ at SZA of 90°, similar to the detection limits of earlier instruments, is achieved but at a substantially smaller spatial resolution. Thus TROPOMI OClO measurements provide a clear improvement with respect to previous instruments."

*Specific (minor) comments:*

*Page 1, line 5: Should read 'From the measured spectra, highly resolved …'*

The text has been removed according to the suggestion by the Reviewer 1

*Page 1, line 11: Just use OClO since this has been already introduced in the paragraph above.*

After the removal of the text before, OClO now is introduced here for the first time

All comments below are considered as suggested

*Page 1, line 15: '… effects, a higher order …'*

*Page 1, line 21: typo: 'operational'*

*Page 10, line 125: left bracket is missing*

*Page 12, line 250: '… zenith sky …'*

*Page 12, line 252: '… in a fit window of …'*

*Page 12, line 252: replace 'considered' with either 'included' or 'used'*

*Page 17, line 306: '.. are listed: The retrieval …'*

*Page 17, line 309: '.. terms are applied (or used).'*

*Page 17, line 311: '... within the 89 ....'*

*Page 17, line 316: '(Sect. ??)' needs to be fixed*

*Page 17, lines 318-319: '... with the correlation ... has an offset ...'*

*Most pages have sentences where commas are missing but I assume that this will be addressed anyway during the proof-reading phase.*